# Griffin: Towards a Graph-Centric Relational Database Foundation Model

Yanbo Wang [1] [†]  Xiyuan Wang [1]  Quan Gan [2]  Minjie Wang [2]  Qibin Yang [1] [†]  David Wipf [2]  Muhan Zhang [1]

## Abstract

We introduce Griffin, the first foundation model attempttation designed specifically for Relational Databases (RDBs). Unlike previous smaller models focused on single RDB tasks, Griffin unifies the data encoder and task decoder to handle diverse tasks. Additionally, we enhance the architecture by incorporating a cross-attention module and a novel aggregator. Griffin utilizes pretraining on both single-table and RDB datasets, employing advanced encoders for categorical, numerical, and metadata features, along with innovative components such as cross-attention modules and enhanced message-passing neural networks (MPNNs) to capture the complexities of relational data. Evaluated on large-scale, heterogeneous, and temporal graphs extracted from RDBs across various domains (spanning over 150 million nodes), Griffin demonstrates superior or comparable performance to individually trained models, excels in low-data scenarios, and shows strong transferability with similarity and diversity in pretraining across new datasets and tasks, highlighting its potential as a universally applicable foundation model for RDBs. Code available at github.com/yanxwb/Griffin.

## 1. Introduction

Foundation models have revolutionized domains such as natural language (Brown, 2020; Touvron et al., 2023), vision (Wang et al., 2023; Yuan et al., 2021), tabular data (Zhang et al., 2023b; Yang et al., 2024), and graphs (Liu et al., 2023; Tang et al., 2024), offering scalable and generalized frameworks for diverse tasks. However, foundation models for RDBs—where tables are interconnected through complex relationships—remain underexplored, despite the practical importance and widespread use of RDBs.

An RDB foundation model is defined as a predictive model that works across diverse RDBs with varying *sizes*, *schemas*, and *domains*. Integrating RDBs into the foundation model paradigm presents challenges, including structural complexity, lack of processing pipelines, and unique computational patterns that differ from single-table data. While traditional machine learning and deep learning methods could handle single-table data pretty well (Chen & Guestrin, 2016; Ke et al., 2017; Prokhorenkova et al., 2018; Huang et al., 2020; Arik & Pfister, 2021; Gorishniy et al., 2021), they fail to address RDB-specific complexities. On the other hand, flattening relational data into a single table often results in significant information loss (Cvitkovic, 2020; Chepurko et al., 2020). Recently, graph-based methods using Graph Neural Networks (GNNs) (Kanter & Veeramachaneni, 2015; Cvitkovic, 2020; Bai et al., 2021; Zhang et al., 2023a; Wang et al., 2024; Robinson et al., 2024) attempt to capture these relationships but are limited to task-specific models rather than proposing a universal RDB foundation model.

In this work, we introduce Griffin, a **G**raph-centric **R**elat**I**onal database **F**oundat**IoN** model attempttation, which integrates pretraining on both single-table and RDB datasets. Griffin is designed for broad generalization across diverse tasks and demonstrates superior performance over task-specific GNN approaches through several key innovations.

The first challenge is how to handle different task types (classification and regression) and different RDBs with disparate input feature spaces (categorical, numerical, textual, etc. of diverse semantic meanings). Griffin **unifies input data encoders and task decoders**. Unlike earlier methods that apply a single embedding layer to all categorical features and directly input raw numerical values, Griffin uses a pretrained text encoder for categorical inputs and a pretrained float encoder for numerical ones. It also incorporates RDB metadata, including table names, column names, and edge types, to distinguish tasks and capture the structure that connects them. In contrast to prior work that uses separate prediction heads for each task, Griffin applies a shared float decoder (pretrained jointly with the float encoder) for all regression tasks, and a unified classification head that integrates the text embeddings of target categories. This allows Griffin to manage classification tasks with varying numbers of categories and regression tasks with different ranges using a consistent architecture.

---

[†]Work done during internship at Amazon. [1]Institute for Artificial Intelligence, Peking University. [2]Amazon Web Services. Correspondence to: Muhan Zhang <muhan@pku.edu.cn>.

*Proceedings of the 42nd International Conference on Machine Learning*, Vancouver, Canada. PMLR 267, 2025. Copyright 2025 by the author(s).

Another key design consideration is selecting an effective GNN architecture for RDBs. Griffin addresses this by incorporating a **cross-attention module** that flexibly gathers information from cells within a row (treated as a node in the graph), mitigating the loss introduced by mean aggregation in standard GNNs. Additionally, Griffin enhances its **message passing neural network (MPNN)** by performing intra-relation aggregation before merging features across relation types.

A further challenge involves leveraging large-scale data for training. To this end, we constructed a diverse and extensive dataset collection for both single-table and RDB tasks and developed a multi-stage pretraining and fine-tuning pipeline. Griffin is initially pretrained on single-table datasets using a random masked cell completion task that does not require labeled data. This is followed by joint supervised fine-tuning (SFT) on realistic tasks from both single-table and RDB datasets, and finally by task-specific fine-tuning and evaluation on each downstream RDB task. In total, the pretraining and SFT phases covered over 150 million nodes (rows), enabling the formation of large, heterogeneous, and temporal graphs across various domains to support the large model development.

To assess the effectiveness of the model design and the impact of pretraining, Griffin was evaluated on two recent graph-centric RDB benchmarks: 4DBInfer (Wang et al., 2024) and RelBench (Robinson et al., 2024). The evaluation led to the following key findings: (1) Even without pretraining, the Griffin architecture achieves significant improvements on downstream tasks, demonstrating the advantages of its advanced design. (2) Pretraining and SFT solely on single-table datasets enables Griffin to outperforms its non-pretrained counterpart. (3) With SFT on RDBs of similarity and diversity to downsteam tasks, Griffin achieves even better results, particularly in scenarios with limited downstream task samples, highlighting its potential as a foundation model capable of transferring to downstream tasks with limited downstream supervision.

In summary, Griffin represents a significant step forward in the development of foundation models for RDBs by combining robust generalization capabilities with architectural innovations that address the complexities of relational data.

## 2. Preliminary

A *Relational Database (RDB)* is formally defined as a collection of tables, denoted by $\mathcal{R} = \{\boldsymbol{T}^k\}_{k=1}^K$, where $K$ represents the total number of tables, and each table $\boldsymbol{T}^k$ is structured as a matrix with $N_k$ rows (instances) and $M_k$ columns (features). The individual entry in the $i$-th row and $j$-th column of table $\boldsymbol{T}^k$ is represented by $\boldsymbol{T}_{i,j}^k$. In our setting, these entries can take various forms, including numerical values, categorical values, text, or hash values used for indexing.

A key characteristic of an RDB is the relationships between tables, which are defined by *Primary Keys (PKs)* and *Foreign Keys (FKs)*. A PK is a column within a table that uniquely identifies each row, while a FK is a column in another table that references the PK, thereby inheriting values from the corresponding PK. Let $R$ denote the total number of PK-FK pairs across all tables.

The *heterogeneous graph* derived from an RDB is formally defined as $\mathcal{G} = (\{\mathcal{V}^k\}_{k=1}^K, \{\mathcal{E}^r\}_{r=1}^R)$, where the node set $\mathcal{V}^k$ of type $k$ is constructed from the rows of table $\boldsymbol{T}^k$, with each node corresponding to a row in the table. The feature vector of each node is the corresponding row in the table, and the edge set $\mathcal{E}^r$ of type $r$ is constructed from the $r$-th PK-FK pair, which connects rows from the table with the FK to the referenced rows in the table with the PK. In this graph-based representation, almost all hash values used for indexing (such as those for PKs and FKs) are already encoded in the edge connection patterns. Therefore, we use only on the numerical, categorical, and textual features for the node attributes.

A common approach in tabular data prediction is missing value prediction, where the goal is to infer a missing cell value using information from the same table or related tables. In many real-world applications, table rows are associated with a *timestamp*, denoted as $t_i^k$, and the number of rows can be very large. To maintain temporal causality and reduce memory usage, only rows with timestamps earlier than that of the target row are allowed for use in prediction. In the graph-based formulation of this task, the problem is rephrased as sampling a rooted subgraph that includes only nodes (rows) with earlier timestamps than the queried node.

Let the target column value be represented as $\boldsymbol{T}_{i',j'}^{k'}$, associated with the node $\mathcal{V}_{i'}^k$. This node serves as the root of a temporal, rooted computation subgraph: $\mathcal{T}^{(L)} = (\{\mathcal{V}^{(l)}\}_{l=0}^L, \{\mathcal{E}^{(l)}\}_{l=0}^{L-1})$, where $\mathcal{V}^{(0)} = \{(\mathcal{V}_{i'}^k \setminus \boldsymbol{T}_{i',j'}^{k'})\}$ is the root node, which does not include the target column's feature. The set $\mathcal{V}^{(l)}$ contains nodes at hop $l$ from the root and only includes nodes satisfying that $t < t_{i'}^{k'}$.

Each edge set $\mathcal{E}^{(l)}$ include edges connecting nodes in $\mathcal{V}^{(l+1)}$ to parent nodes in $\mathcal{V}^{(l)}$, and satisfies: $\forall \mathcal{V}_p^{(l+1)} \in \mathcal{V}^{(l+1)}, \exists \mathcal{V}_q^{(l)} \in \mathcal{V}^{(l)}$ with $(\mathcal{V}_p^{(l+1)}, \mathcal{V}_q^{(l)}) \in \mathcal{E}^{(l)}$.

Through this transformation, the RDB task of predicting a column value from multiple related tables is converted into a graph-based prediction problem over a temporally constrained rooted subgraph, defined as follows:

- **Input:** A sampled rooted subgraph $\mathcal{T}^{(L)}$ constructed for the target column $\boldsymbol{T}_{i',j'}^{k'}$.

- **Output:** Predicted value of the target column $\boldsymbol{T}_{i',j'}^{k'}$.

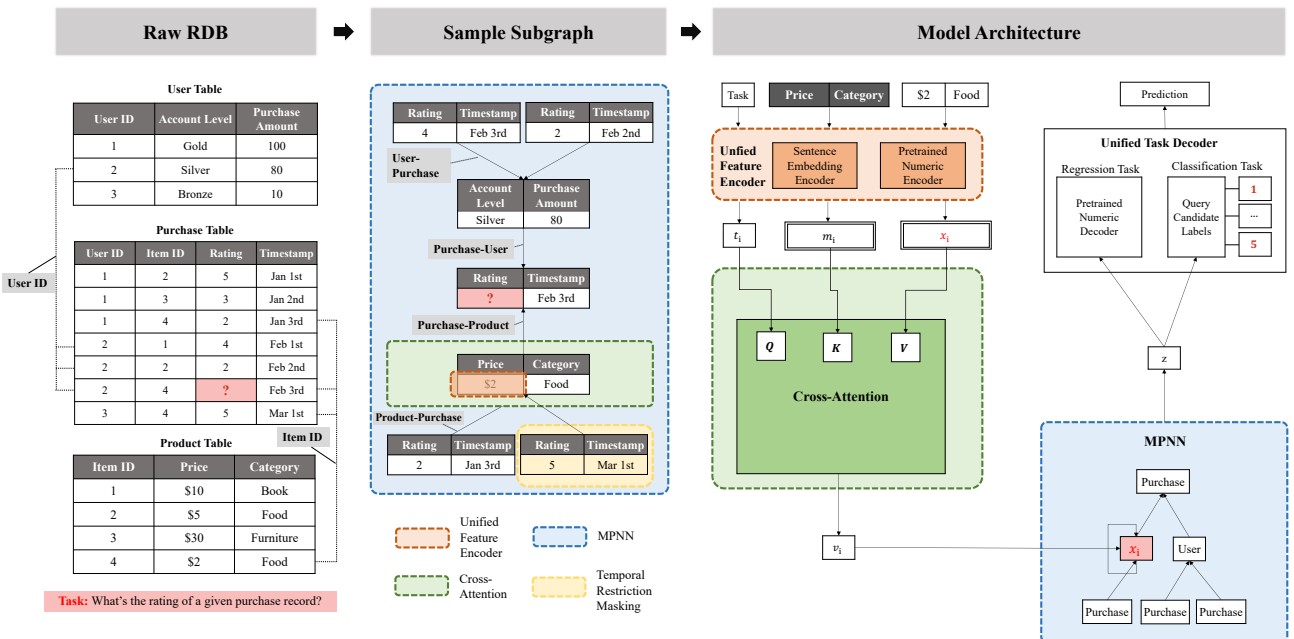

Figure 1: **Overview of the Griffin Model Framework**. The framework first transforms RDBs into a graph structure by representing each row as a node and using primary key–foreign key relationships as edges. Given a target column, a temporally constrained subgraph is sampled and processed using a unified encoder module before being passed to a MPNN. Finally, the unified task decoders generate predictions based on whether the task is classification or regression.

## 3. Model Design

In this section, we present the model design of Griffin, as illustrated in Figure 1. The framework consists of three main components for processing a sampled subgraph and generating prediction values: a unified data encoder, a MPNN tailored for RDBs, and a unified task decoder.

### 3.1. Unified Data Encoder

A core innovation of Griffin is the unification of different RDB input data. Previous RDB models typically use separate embedding layers for categorical features and direct numerical inputs, which makes them difficult to generalize to new data. In contrast, Griffin handles all categorical and text features using a pre-trained text encoder, while numerical features are normalized and processed with a pre-trained float encoder. This approach ensures that input distributions are more consistent across tasks. Additionally, Griffin uses RDB metadata and task-specific information to create task embeddings, allowing the same model to perform different tasks based on the task embedding provided at the input.

**Categorical and Textual Feature** For categorical features, we first convert them into text representations using the metadata from the RDB. Both categorical and text features are then passed through a single, pre-trained text encoder (Nussbaum et al., 2024). Each feature (or "cell") is encoded into a fixed-length vector, which captures rich semantic information. Cosine similarity between these vectors allows us to measure the similarity between different texts.

**Numerical Feature** For numerical data, to avoid issues with extreme values, we first apply a quantile normalizer (Bolstad et al., 2003) to standardize the numerical values, transforming the distribution to normal. We then use a pre-trained Multi-Layer Perceptron (MLP) to convert the normalized values into the same $d$-dim embedding vectors. To train the MLP, we sample $x$ from a normal distribution:

$$w = \text{ENC}(x) \in \mathbb{R}^d, \quad y = \text{DEC}(w) \in \mathbb{R}, \quad (1)$$

where ENC encodes the float into an embedding, and DEC decodes the embedding back to a float value. The model is trained using L1 loss ($|y - x|$), and we apply LayerNorm (without affine weights) to the encoder's output to prevent collapse. After pretraining, the encoder and decoder are fixed and do not participate in the training of the Griffin model. During inference, the numerical input is first normalized, then passed through the encoder.

**Metadata Information** Griffin also incorporates flexible encoding of RDB metadata, such as table names, column names, and edge types. This metadata is encoded with the text encoder to provide additional node and edge features.

**Task Representations** These unified input encoders map inputs from different tasks to the same space. However, the model also needs to account for the syntactic differences between tasks. For example, if two missing cells in the same row are predicted without additional task-specific embeddings, the model's input for both tasks would be identical,

leading to the same output representation and poor expressivity. To address this, we introduce a task embedding. This embedding is generated by the text encoder using the column name of the cell to be predicted as input, allowing the model to produce distinct embeddings for different tasks.

With all components unified across different types of data, the sampled subgraph is extended as follows:

- **Input:** A sampled rooted subgraph $\mathcal{T}^{(H)}$ constructed for the target column $\boldsymbol{T}_{i',j'}^{k'}$.
- **Output:** An enriched rooted subgraph in which each node $i$ is associated with a feature tensor $x_i \in \mathbb{R}^{L_i \times d}$ and a metadata tensor $m_i \in \mathbb{R}^{L_i \times d}$, where $L_i$ is the number of cells in the node (which may vary between nodes). Each edge $(i, j)$ of relation type $r$ carries a relation-specific metadata vector $e_r \in \mathbb{R}^d$. Additionally, a task embedding vector $t \in \mathbb{R}^d$ is provided to represent task-specific information.

## 3.2. MPNN Architecture

The MPNN of Griffin is composed of multiple layers, each containing two key components: a cross-attention module that extracts information from node features and a message-passing module that facilitates information exchange between nodes. The intermediate embedding of node $i$ across layers is maintained as $u_i$, while the final output of the model is the representation of the target node, denoted as $z$.

**Cross Attention Module** The node feature vector $x_i \in \mathbb{R}^{L_i \times d}$ presents three challenges:

- The number of cells $L_i$ varies across nodes, so the encoder must handle variable-length inputs.
- $x_i$ contains rich information, some of which may not be relevant to the task. The encoder must selectively focus on task-relevant information.
- In RDB data, the column order is meaningless, so the encoder should be invariant to column permutation for better generalization.

To address these, Griffin introduces a cross-attention module that allows the model to selectively focus on relevant information from individual cells within a row (treated as a node in a graph). This enables the model to capture interactions between columns and rows, modeling complex dependencies critical for relational data analysis.

Each row in a table is represented as an attention-based aggregation of its column data:

$$v_i^l = \text{Attention}_l \left( \text{QMLP}_l(u_i, t), m_i, x_i \right), \qquad (2)$$

where $l$ is the layer index, $v_i^l$ is the output of the attention mechanism for node $i$, and $\text{QMLP}_l$ is an MLP that takes the node representation $u_i \in \mathbb{R}^d$ and task representation $t \in \mathbb{R}^d$ as inputs to produce the query for cross-attention.

The keys for cross-attention are the metadata $m_i \in \mathbb{R}^{L_i \times d}$ (column names) of node $i$, and the values are $x_i \in \mathbb{R}^{L_i \times d}$, the input node features. The result, $v_i^l$, is added to $u_i$ to update the node representation. This cross-attention module overcomes the information loss typically seen in traditional GNNs, which often aggregate different columns using simple methods like mean aggregation. By focusing on specific cells in a row and attending to their contextual relationships, the module improves the model's ability to extract nuanced information, enhancing task performance.

**Hierarchical Aggregation** Along with the cross-attention module, Griffin enhances its MPNN to reduce information loss. Instead of aggregating all neighbors uniformly, Griffin first aggregates information within each relation type and then combines features across different relations. This hierarchical aggregation helps preserve the structure of relational data by ensuring that information is aggregated within each relation (e.g., a specific table or type of relationship) before being combined across multiple relations. This approach prevents the loss of important relational context and helps the model learn more informative representations.

Specifically, in cross-table modeling, Griffin uses a temporal heterogeneous graph representation of the RDB, where rows are modeled as nodes. These node embeddings are propagated and updated via a GNN. The embedding for a node $i$ at the $l$-th layer, denoted as $h_i^l$, is updated as follows:

$$h_i^{r,l} = \text{Mean}_l \left( \text{AMLP}_l(u_j) \mid (i, j) \in \mathcal{E}^r \right), \qquad (3)$$

$$h_i^l = \text{Max}_l \left( h_i^{r,l} \odot e_r \mid r \in R \right), \qquad (4)$$

where AMLP is an MLP that transforms the aggregated node representations. First, the representations of neighboring nodes are averaged within each relation. Then, the maximum aggregation is applied across all relations. This step ensures that the representations from each relation are not overwhelmed by others, which can be problematic when the number of neighbors across relations is unstable. $h_i$ is further added to $u_i$ to update node representations.

The subgraph is encoded to one vector by MPNN:

- **Input:** The enriched rooted subgraph and task vector.
- **Output:** A fixed-length vector $z \in \mathbb{R}^d$.

## 3.3. Unified Task Decoder

Given a fixed-length embedding from the MPNN output, we apply a single task decoder per task type.

**Classification tasks** We directly use the text embeddings of the target labels as the classification head. For example, when predicting the value of the $(i, j)$-th cell in a table, let $z_1, z_2, \ldots, z_c \in \mathbb{R}^d$ denote the text embeddings of all candidate categories, and let $z \in \mathbb{R}^d$ be the output vector

from Griffin. The prediction probability distribution is:

$$\text{softmax}([\langle z, z_i \rangle \mid i = 1, 2, \ldots, c]), \quad (5)$$

where the logit for each category is obtained by the inner product between the output vector $z$ and the corresponding category embedding $z_i$.

**Regression tasks** The output vector is passed through a pretrained number decoder, denoted as DEC, to produce the predicted value. The final output can then be denormalized according to the specifications of the downstream task.

Different tasks may share similar label embeddings or number distributions, allowing the model to better capture task-specific characteristics and adapt to new tasks. Given the decoder design, the final prediction step is defined as:

- **Input:** A fixed-length vector $z \in \mathbb{R}^d$.
- **Output:** The predicted value for the target column $\boldsymbol{T}_{i',j'}^{k'}$.

## 4. Training Pipeline

In this section, we describe the training pipeline of Griffin, which consists of pretraining and downstream task fine-tuning stages. The pretraining phase includes two components: Completion Pretraining and Joint Supervised Fine-Tuning (SFT). Both are designed to remain independent of the downstream tasks to avoid task-specific bias. The final stage involves task-specific fine-tuning, where Griffin is adapted to individual downstream tasks.

### 4.1. Pretraining

Griffin is pretrained on a diverse set of datasets to ensure effective generalization across various RDB tasks. The pretraining process has two main components:

- **Single-Table Datasets**: These are used to train the model on tasks involving individual tables, providing a foundational understanding of tabular data.
- **RDB Datasets**: Griffin is also pretrained on large-scale, heterogeneous, and temporal graphs derived from multiple RDB domains. These graphs capture data structures.

By using both single-table and relational data, Griffin learns to generalize across different types of RDBs, making it adaptable to a wide variety of tasks. To fully use rich sourced datasets, we include two stages for pretraining: **completion pretraining** and **joint SFT**.

**Completion Pretraining** We first use a completion task similar to language modeling but adapted for the tabular domain. The model learns to predict masked values within a row based on the remaining data. For a given row where one column is randomly masked, a column-invariant row encoder is used to generate the masked row's embedding, which is used to predict the masked value.

Formally, for a row $\boldsymbol{T}_{i,:}^k$ with a target column $j'$ to be predicted, the pretraining objective is defined as:

$$\text{loss} = 1 - \cos\left(\text{Model}_\theta\left(\boldsymbol{T}_{i,:\backslash j'}^k\right), \text{Encoder}\left(\boldsymbol{T}_{i,j'}^k\right)\right), \quad (6)$$

where $\text{Model}_\theta\left(\boldsymbol{T}_{i,:\backslash j'}^k\right)$ generates the row embedding and $\text{Encoder}\left(\boldsymbol{T}_{i,j'}^k\right)$ provides the true embedding for the masked column. The objective minimizes the cosine distance between the predicted and true embeddings.

**Joint Supervised Fine-Tuning** Following completion pretraining, Griffin is jointly fine-tuned on selected realistic tasks to align it more closely with real-world tabular tasks. This stage utilizes both labeled single-table datasets and carefully selected RDB datasets, ensuring no data leakage into downstream evaluations.

The fine-tuning process optimizes the model for a set of related tasks, leveraging pretrained knowledge while adapting to the specific needs of tabular tasks. Griffin's task modeling framework, which supports both classification and regression in a graph-based RDB representation, employs a unified decoder (as in Section 3.3) to map output embeddings to task predictions. Cross-entropy loss is used for classification tasks, while L2 loss is for regression tasks.

### 4.2. Downstream Task Fine-Tuning

After completing pretraining and joint SFT, Griffin is fine-tuned on each individual downstream task for evaluation. This process follows the specific pipeline requirements of each benchmark to ensure a fair comparison with baselines.

### 4.3. Model Variants Considered

In practice, we consider three model variants based on the datasets used during training:

**Griffin-unpretrained** refers to the model without any pretraining. This variant differs from other GNN baselines only in its architectural design, with no exposure to external data prior to downstream training, aiming to reveal the advantages of Griffin model design.

**Griffin-pretrained** refers to the model pretrained exclusively on single-table datasets. We use a single pretrained checkpoint for fine-tuning across all downstream tasks. This checkpoint is strictly disjoint from any downstream tasks, ensuring there is no data leakage. This configuration isolates the effect of pretraining and highlights the potential for building a general-purpose foundation model for RDBs.

**Griffin-RDB-SFT** is trained on a combination of single-table and RDB datasets. Since RDB datasets are used for both pretraining and downstream fine-tuning, we maintain multiple checkpoints, each trained on subsets of the RDB data that are non-overlapping with the downstream tasks. This setup also enables us to investigate the transferability of knowledge across different RDBs.

# 5. Related Work

## 5.1. Tabular Predictive Tasks

Tabular predictive tasks involve learning to estimate missing or target values in structured tables. These tasks typically include classification and regression, using available features from the same table or from related tables. Models are trained to capture statistical patterns within rows, across columns, and across multiple tables when relational data is present. Our model focuses specifically on this type of task.

**Single Table Models**  Research on single table data has evolved through various approaches. Traditional methods, such as XGBoost (Chen & Guestrin, 2016), LightGBM (Ke et al., 2017), and CatBoost (Prokhorenkova et al., 2018), have been widely adopted due to their scalability and strong performance on structured data. More recently, transformer-based methods like TabTransformer (Huang et al., 2020), TabNet (Arik & Pfister, 2021), FT-Transformer (Gorishniy et al., 2021), and SAINT (Somepalli et al., 2021) have leveraged attention mechanisms to capture complex relationships within rows and columns. Additionally, graph-based methods such as GRAPE (You et al., 2020), TabularNet (Du et al., 2021), TabGNN (Guo et al., 2021), and CARTE (Kim et al., 2024) represent tabular data as graphs, incorporating multiplex and hypergraph structures to model interactions between rows and columns more effectively. Other works have explored improved encoding strategies for numerical features (Gorishniy et al., 2022; Yarullin & Isaev, 2023), while some have highlighted the benefits of incorporating nearest-neighbor information (Gorishniy et al., 2023; Ye et al., 2025). Although these models enhance feature interaction modeling, they primarily focus on single-table datasets and typically fail to model relational dependencies across multiple tables.

**RDB Models**  RDBs extend the concept of single-table models by incorporating multiple interrelated tables, requiring models to capture both intra- and inter-table relationships. Early approaches, such as DFS (Kanter & Veeramachaneni, 2015) and RDBTOGRAPH(Cvitkovic, 2020), attempt to flatten RDBs into a single table or apply GNNs to model relationships between tables. Other works, like ATJ-Net (Bai et al., 2021) and KEN (Cvetkov-Iliev et al., 2023), use hypergraphs and knowledge graphs to model inter-table dependencies, while GFS (Zhang et al., 2023a) integrates differentiable single-table models as embedding functions to preserve table structures. Some methods that convert structured data into unstructured embeddings can still retain structural information (Grover & Leskovec, 2016), such as EmbDi (Cappuzzo et al., 2020) and RDF2Vec (Ristoski & Paulheim, 2016). As RDB tasks have attracted increasing attention (Fey et al., 2024), more comprehensive benchmarks and toolboxes have emerged. For example, 4DBInfer (Wang et al., 2024), RelBench (Robinson et al., 2024; Fey et al.,

2023), and PytorchFrame (Hu et al., 2024) propose complete pipelines for converting RDBs into graph structures that can be used for GNN-based models. More recent efforts (Yuan et al., 2024; Chen et al., 2025) aim to design more expressive GNN architectures for relational data. These models perform well on individual RDB tasks, whereas Griffin is designed towards a foundation model that aims to generalize across a wide range of relational tasks.

## 5.2. Table QA Tasks

Table question answering (QA) tasks focus on answering natural language queries by reasoning over tabular data. Given a question and a table (or a set of tables), the model must interpret the query, identify relevant cells, and either extract or compute the correct answer, or generate an executable SQL query. These tasks require both natural language understanding and structured data reasoning. TaPas (Herzig et al., 2020) enhances BERT with a table-aware encoder. Tapex (Liu et al., 2021) explores learning a neural SQL executor. OmniTab (Jiang et al., 2022) introduces pretraining using both synthetic and natural datasets. TableGPT2 (Su et al., 2024) treats tabular data as a distinct modality for building general-purpose models. Numerous benchmarks have been proposed for comprehensive evaluation (Yu et al., 2018; Lei et al., 2024; Chen et al., 2019; Wu et al., 2024; Li et al., 2023; Qiu et al., 2024).

## 5.3. Foundation Models for Predictive Tasks

**Graph Foundation Models (GFMs)**  aim to pretrain large models that generalize across multiple graph datasets and tasks. Many GFMs, such as OFA (Liu et al., 2023) and Tape (He et al., 2023), integrate Large Language Models (LLMs) to enhance feature spaces or assist in training GNNs. Other methods, like UniGraph (He & Hooi, 2024), adapt graph data for better LLM integration. While some GFMs, such as GraphText (Zhao et al., 2023), convert graph structures into language-like representations for processing by LLMs, others focus on novel GNN architectures, such as GraphAny (Zhao et al., 2024). Griffin builds on the GFM paradigm but adapts it to RDBs by pretraining on both single-table and multi-table data, incorporating advanced tabular-specific data encoders and graph-based components such as cross-attention to model table meta-information, making Griffin more suitable for RDBs compared to GFMs.

**Tabular Foundation Models (TFMs)**  aim to generalize across tabular data, often leveraging transformer-based architectures. Models such as TaBERT (Yin et al., 2020) and TabLLM (Hegselmann et al., 2023) integrate text and tabular data to enhance table structure understanding, while TransTab (Wang & Sun, 2022) and XTab (Zhu et al., 2023) explore transfer learning across tables with varying column structures. UniTabE (Yang et al., 2024) and TPBerta (Yan et al., 2024) employ specialized tabular encoders to better

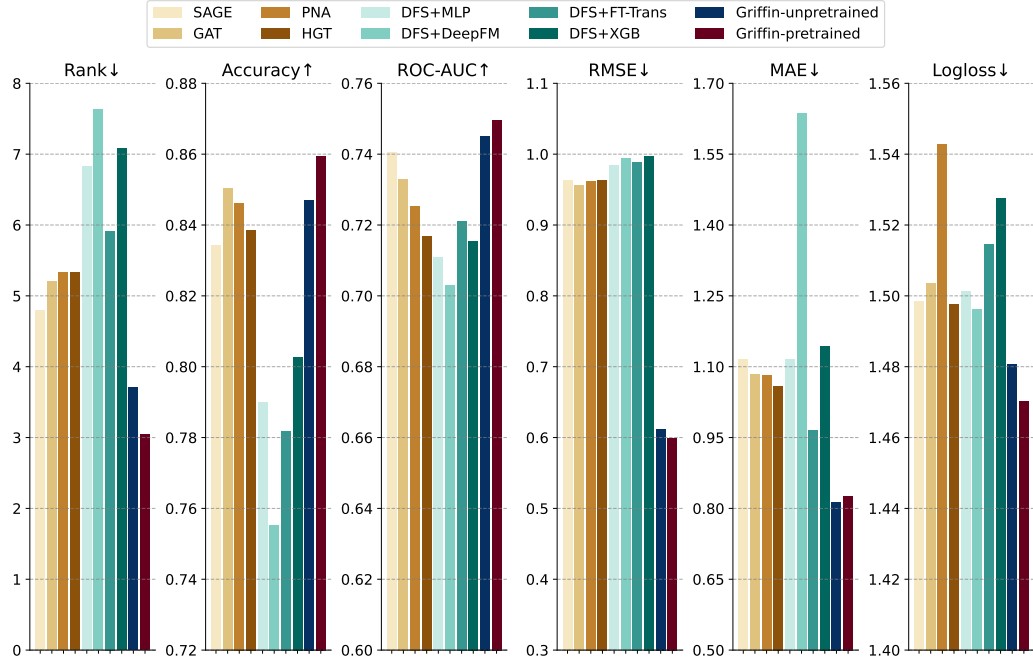

Figure 2: **Performance Comparison of Fully Fine-Tuned Models on Individual Tasks.** This figure compares the performance of four GNN baselines, four single-table baselines with DFS, and two Griffin variants, each fine-tuned on individual tasks. The leftmost subfigure presents the average rank across all tasks. The remaining subfigures group tasks by evaluation metric, with results averaged accordingly. All values are positive; higher values indicate better performance for Accuracy and ROC-AUC, while lower values are better for left ones.

align transformers with tabular formats. TabPFN (Hollmann et al., 2022; 2025) takes a different approach by avoiding the use of text models and instead pretraining on a large number of synthetic datasets. It achieves strong performance in few-shot settings. However, these models primarily focus on single-table data and lack mechanisms to capture inter-table relationships in RDBs. While Griffin incorporates transformer-based and tabular techniques, it extends beyond existing TFMs by explicitly modeling relational structures across multiple tables, addressing the complexities in RDBs.

## 6. Experiments

In this section, we aim to address the following questions:

**Q1:** Can Griffin, with its advanced design, outperform existing models under the same training settings?

**Q2:** Can utilizing a single pretrained checkpoint universally enhance predictive performance?

**Q3:** Can joint SFT with RDB improve transferability, and under what conditions does it provide the most benefit?

### 6.1. Experimental Setup

The experimental setup is designed to evaluate Griffin across diverse tasks and datasets, leveraging the pretraining and fine-tuning pipeline described in Section 4.

**Datasets** The selected datasets include both single-table and RDB datasets, with details provided in Appendix A.

- **Single-Table Datasets**: Over 200 datasets were curated from TPBerta (Yan et al., 2024) and CARTE (Kim et al., 2024), comprising approximately 10 million rows. These datasets were used for completion pretraining, enabling scalable learning without human-labeled data. Only 50 datasets contained labels for joint SFT. While additional large-scale datasets from diverse domains were collected, they were excluded from pretraining for two key reasons: (1) many were subsets of RDBs, making single-table pretraining ineffective, and (2) their distributions diverged significantly from downstream RDBs.

- **RDB Datasets**: We sourced large-scale temporal RDBs from two leading benchmarks, 4DBInfer (Wang et al., 2024) and RelBench (Robinson et al., 2024), covering a wide range of domains, scales, and tasks. A total of 24 tasks were selected for SFT and downstream evaluation.

**Baselines** To ensure a fair comparison across benchmarks, we standardized evaluation-related settings, which led to certain modifications in the reported results. These adjustments include aligning preprocessing steps, normalization strategies, and other evaluation procedures. As a result, some baseline results may differ from those originally reported in the respective benchmarks. We include four GNN baselines: SAGE, GAT, PNA, and HGT. Additionally, we evaluate four single-table models enhanced with the Deep Feature Synthesis (DFS) method (Kanter & Veeramachaneni, 2015) to incorporate multi-table information. For evaluations that involve only single-table data without any relational context,

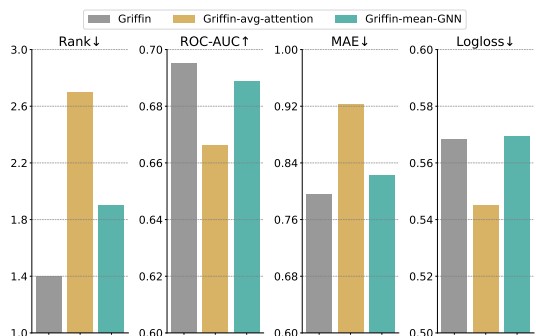

Figure 3: **Ablation Study on Different Model Design Choices.** This figure compares the performance of Griffin-unpretrained and two ablated variants, with cross-attention and max-aggregation removed, respectively. The leftmost subfigure presents the average rank across all tasks. The remaining subfigures group tasks by evaluation metric, with results averaged accordingly. All values are positive; higher values indicate better performance for ROC-AUC, while lower values are better for left ones.

we recommend referring to the original experimental results reported in 4DBInfer and RelBench, which have already demonstrated significantly weaker performance in the absence of multi-table information. Further details on these modifications and baseline configurations are provided in Appendix B.

**Hyperparameters and Training** Griffin was trained with fixed hyperparameters across all experiments to ensure robustness. During pretraining, all single-table datasets were used for completion pretraining, while subsets of single-table and RDB datasets were selected for joint SFT based on specific experimental objectives. Further details about model settings are provided in Appendix C.

## 6.2. Reply to Q1: Meta-Information and Advanced Architecture Design Enhance Model Performance

Figure 2 presents the performance comparison of different models fully fine-tuned on individual tasks. Griffin-unpretrained outperforms all other models in average rank and demonstrates significant improvements.

To analyze the impact of key design choices, we conducted an ablation study on the cross-attention module and aggregation functions in MPNN, with results presented in Figure 3. Replacing these components with a plain average of column features and a mean-only aggregator for both intra-type and inter-type nodes results in a significant performance drop.

## 6.3. Reply to Q2: Single-Table Pretraining Universally Enhances Model Performance

To answer Q2, we introduce Griffin-Pretrained, a variant of Griffin that undergoes a pretraining stage before task-specific fine-tuning. The pretraining process consists of completion pretraining and joint SFT, both conducted exclusively on single-tabular datasets. Notably, no RDB datasets are included in pretraining, ensuring no data leakage while demonstrating the adaptability of the pretraining framework across different domains.

Figure 2 presents the performance comparison between Griffin-Pretrained and Griffin-unpretrained without pretraining across multiple tasks. The results show that Griffin-Pretrained outperforms its non-pretrained counterpart, validating the universal benefits of pretraining. These findings confirm that pretraining a single checkpoint on diverse single-tabular datasets can significantly improve predictive performance, even for tasks in RDBs, despite the absence of RDB-specific data during pretraining.

## 6.4. Reply to Q3: Joint SFT with RDB Enhances Transferability, Driven by Similarity or Diversity

To analyze the factors influencing transferability, we propose two key hypotheses, building on recent research on transferability (Ehrig et al., 2024).

- **Similarity**. Pretraining on datasets similar to the downstream task improves transferability by providing aligned feature representations and task structures.
- **Diversity**. Pretraining on a broader and more diverse set of datasets enhances transferability by improving the model's ability to generalize across different domains.

To test these hypotheses, we categorized datasets into two broad domains: commerce and others, each containing a diverse set of tasks. We further split each domain into two subsets, leading to four final groups: Commerce-1, Commerce-2, Others-1, and Others-2.

- **Commerce**: Commerce-1 and Commerce-2 originate from e-commerce datasets, covering tasks such as user churn prediction and purchase rate estimation. These datasets are highly similar.
- **Others**: Others-1 and Others-2, despite belonging to the same broad domain, exhibit significant internal diversity due to their inclusion of sports, social networks, flight records, and clinical data, leading to varying distributions.

To evaluate transferability, we performed joint SFT on one group and tested it on each task of another group using limited-sample fine-tuning, making transfer effects clearly visible. To ensure robustness, each task was evaluated across five different random seeds for split selection. We compared two types of models to assess transferability:

- No-Pretrain Model – Only trained on downstream tasks.
- Griffin-RDB-SFT – Griffin pretrained on both single-tabular and selected RDB datasets.

The results, presented in Figure 4, provide insights into how similarity and diversity influence transfer learning.

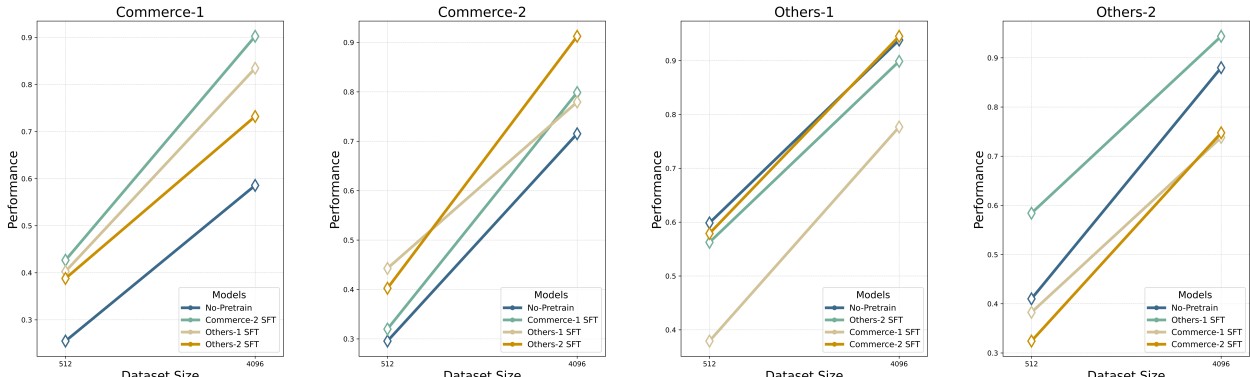

Figure 4: **Evaluating Transferability Across Different SFT Domains.** This figure compares the impact of different SFT strategies on transferability. Each subfigure presents four models: a no-pretraining baseline and three models pretrained on single-table data followed by SFT on different domains. By comparing performance relative to the no-pretraining baseline, we observe positive transfer effects when the SFT and downstream tasks are similar, as seen in commerce-to-commerce settings. Additionally, SFT on the more diverse "Others" group improves performance on the "Commerce" domain.

**Impact of Similarity on Transferability** To evaluate the role of similarity, we first analyze transfer performance between the commerce groups. The results show that both Commerce-1 to Commerce-2 and Commerce-2 to Commerce-1 benefit significantly from pretraining. Notably, pretraining on Commerce-2 and transferring to Commerce-1 outperforms all other pretraining domains, indicating that pretraining on highly similar datasets results in stronger transfer benefits. Conversely, when transferring from commerce to non-commerce domains, the lack of similarity leads to poor transfer performance. In these cases, pretraining often underperforms compared to the no-pretrain baseline, resulting in a substantial performance gap. This trend confirms our Similarity Hypothesis—the closer the pretraining dataset is to the downstream task, the stronger the transferability.

**Impact of Diversity on Transferability** Next, we examine the role of diversity in transfer learning. When transferring from others to commerce, pretraining consistently outperforms the no-pretrain model, and in some cases, even surpasses the results of similar-domain commerce-to-commerce transfer. This suggests that diverse pretraining can sometimes be as effective as, or even better than, pretraining on similar datasets.

For transfering between Others-1 and Others-2, where similarity is low, we observe a one-directional transfer benefit—Others-1 effectively transfers to Others-2, but not vice versa. We hypothesize that this is due to greater dataset diversity in Others-1, which provides a broader pretraining foundation that improves generalization when fine-tuned on Others-2. These findings confirm our Diversity Hypothesis—the more diverse the pretraining dataset, the stronger the model's ability to generalize.

Additionally, we conducted further experiments on different

joint SFT strategies, including SFT with limited samples and mixed SFT with single-tabular datasets, as detailed in Appendix D. The results show that full SFT achieves the best performance, reinforcing the benefits of complete pretraining. Furthermore, despite variations in SFT strategies, the observed transferability patterns across domains remain similar. This further validates the robustness of our domain transferability hypothesis and highlights the fundamental role of similarity and diversity in effective transfer learning.

We also conducted few-shot experiments compared with TabPFNv2 combined with DFS as a reference, as detailed in Appendix E. TabPFNv2 (Hollmann et al., 2025) is a powerful single-table model that supports few-shot learning and even outperforms some state-of-the-art models on certain datasets. Although DFS is not ideal for few-shot scenarios and involves substantial preprocessing time, we include it for comparison. The results show that Griffin and TabPFNv2 each excel under different conditions.

## 7. Conclusion

In this work, we proposed a graph-centric foundation model for RDBs that effectively incorporates meta-information, leverages pretraining strategies, and supports diverse downstream tasks. The proposed model demonstrates strong performance across various tasks by unifying single-table and cross-table embeddings using graph-based representations. Through extensive experiments, we addressed three key questions: (1) the model's ability to outperform existing approaches by encoding meta-information, (2) its generalizability across multiple domains with a single pretrained model, and (3) the benefits of similarity and diversity on improving transferability. Our results highlight the importance of aligning model design with the RDB structure and leveraging pretraining to enhance performance across tasks. It provides a robust foundation for advanced RDB research.

## Acknowledgements

This work is supported by the National Key R&D Program of China (2022ZD0160300), National Natural Science Foundation of China (62276003) and Beijing Natural Science Foundation (QY24046).

## Impact Statement

This paper advances the field of Machine Learning for relational databases (RDBs) by exploring pretraining and transferability in structured data. Our findings contribute to improving model generalization and adaptability across domains such as finance, healthcare, and e-commerce, where relational data is widely used.

While our work enhances general predictive performance and transfer learning, it may raises considerations regarding data biases and privacy. The effectiveness of pretraining depends on data quality and representativeness, and improper adaptation could introduce biases in downstream applications. Additionally, working with large-scale relational data may pose privacy risk, particularly in sensitive domains. However, all datasets used in this study are publicly available and have undergone privacy filtering to remove sensitive information, ensuring ethical research practices.

As our research is based entirely on publicly available datasets with no personally identifiable information, we believe no new ethical concerns should be introduced.

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

## A. Datasets

This section details the information of single tabular datasets and RDBs that we use in Griffin.

### A.1. Single Tabular Datasets

We use approximately 200 single tabular datasets from the pretraining datasets of TPBerta (Yan et al., 2024) and datasets of CARTE (Kim et al., 2024) from Hugging Face. The row count distribution of these datasets is shown in Figure 5 and Figure 6, while the column count distribution is shown in Figure 7 and Figure 8. These single tabular datasets cover a wide range of domains, including healthcare, finance and business, social sciences, science and technology, entertainment, media, marketing and so on.

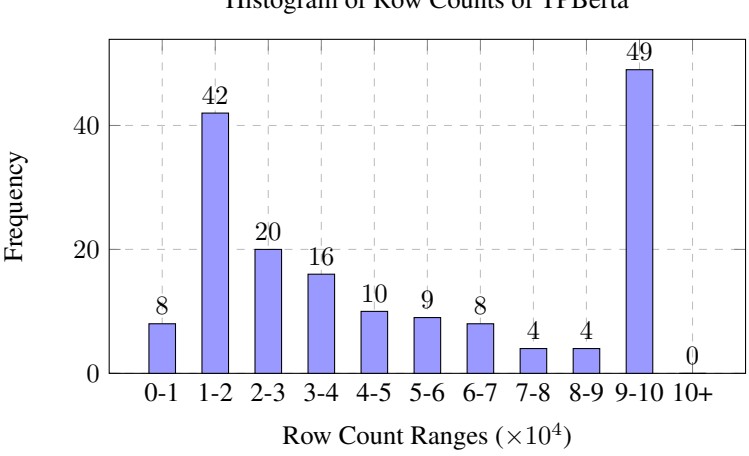

Figure 5: Histogram of row counts of TPBerta

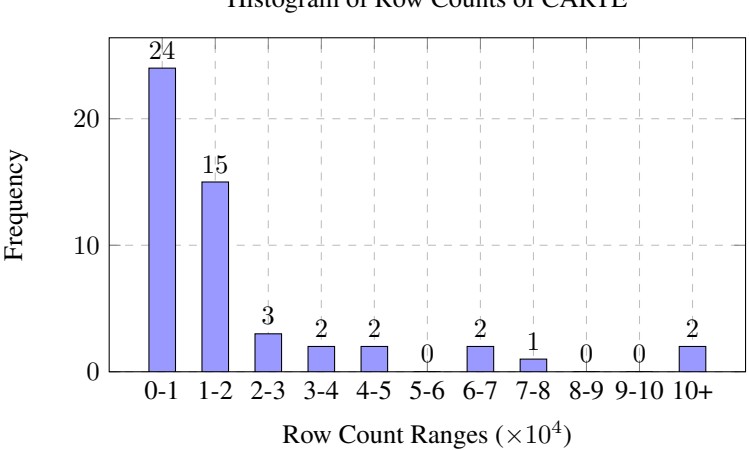

Figure 6: Histogram of row counts of CARTE

### A.2. RDBs

The RDBs that we use are from two benchmarks, 4DBInfer (Wang et al., 2024) and RelBench (Robinson et al., 2024), covering a wide range of domains, scales, and tasks. The detailed information is shown in Table 1.

Histogram of Column Counts of TPBerta

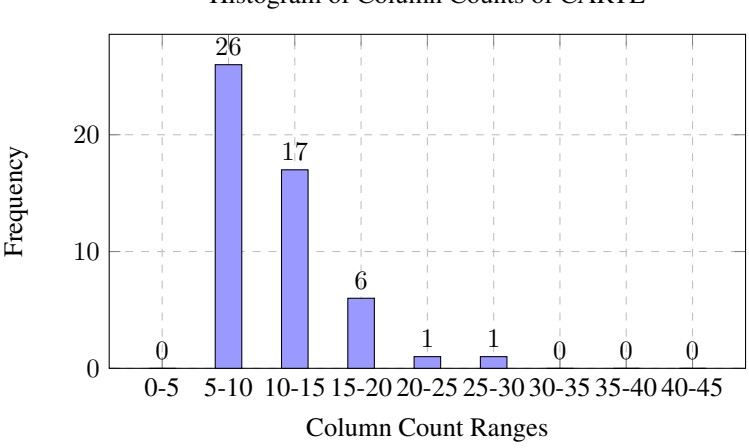

Figure 7: Histogram of column counts of TPBerta

Histogram of Column Counts of CARTE

Figure 8: Histogram of column counts of CARTE

| Dataset | Tables | Columns | Rows |
|---|---|---|---|
| Seznam | 4 | 14 | 2681983 |
| Airbnb | 4 | 34 | 10800000 |
| Amazon | 3 | 15 | 24291489 |
| Diginetica | 5 | 28 | 3672396 |
| Outbrain | 8 | 31 | 4778954 |
| Retailrocket | 3 | 11 | 23033676 |
| Stackexchange | 7 | 49 | 5399818 |
| Virus | 3 | 38 | 145000 |
| Telstra | 5 | 12 | 136000 |
| Talkingdata | 3 | 20 | 36600000 |
| Rel-avito | 8 | 43 | 20679117 |
| Rel-f1 | 9 | 77 | 97606 |
| Rel-hm | 3 | 37 | 33265846 |
| Rel-trial | 15 | 140 | 5852157 |

Table 1: Statistics of relational database datasets.

# B. Baseline Settings

This section details the experimental settings of the baselines used in 4DBInfer, RelBench, and Griffin. While these benchmarks share similarities in graph construction and sampling strategies, they differ in data processing methods and hyperparameter selection, which can impact comparability, particularly for regression tasks.

## B.1. Comparison of 4DBInfer, RelBench, and Griffin Experiment Settings

Table 2 summarizes the key differences among the three baselines.

Table 2: Comparison of Baseline Settings in 4DBInfer, RelBench, and Griffin

| Setting | 4DBInfer | RelBench | Griffin |
|---|---|---|---|
| Graph Construction | R2N / R2NE | R2N | R2N |
| Numerical Processing | Quantile normalization | No normalization | Quantile normalization |
| Time Processing | Categorized and scaled per column | Cyclic encoding | Text description and scaling |
| Text Processing | GloVe embeddings | GloVe embeddings | Sentence model embedding |
| Category Processing | One-hot encoding | One-hot encoding | Text description encoding |
| Sampling Strategy | Fanout is total neighbors across all node types | Fanout per node type | Fanout per node type |
| Hyperparameter Selection | Individually searched per task | Shared across tasks | Shared across tasks |

**Graph Construction and Sampling** All three methods adopt the row-to-node (R2N) approach, with 4DBInfer also extending it to row-to-node-or-edge (R2NE). RelBench and Griffin define fanout per node type, whereas 4DBInfer treats fanout as the total number of neighbors across different node types.

**Data Preprocessing** The benchmarks differ in how they process numerical, temporal, textual, and categorical data:

- Numerical Data: 4DBInfer and Griffin use quantile normalization, while RelBench does not apply specific normalization.

- Temporal Data: 4DBInfer scales time values at the column level, while Griffin incorporates text descriptions alongside scaling. RelBench applies a cyclical encoding method.

- Text Features: 4DBInfer and RelBench use GloVe embeddings (Pennington et al., 2014), while Griffin employs sentence model embeddings for richer representations. Specifically, the model used is Nomic (Nussbaum et al., 2024).

- Categorical Data: 4DBInfer and RelBench use one-hot encoding, whereas Griffin encodes text descriptions instead of static categories.

**Hyperparameter Selection** 4DBInfer performs independent hyperparameter searches per task, while RelBench and Griffin use a shared hyperparameter configuration for consistency across tasks.

## B.2. Metric Adjustments for Regression Tasks

While most differences in preprocessing and sampling strategies reflect model-specific preferences, they generally do not affect comparability across benchmarks. However, for regression tasks, differences in numerical processing can lead to significant variations in results.

4DBInfer applies quantile normalization to all numerical data and predicts normalized values instead of raw targets. To ensure consistency, we adopt a similar approach by implementing a unified number decoder that operates on normalized outputs. In contrast, RelBench's original regression tasks predict raw numerical values, which can range from 0.01 to 100. While our model can make predictions in this format through an additional denormalization step, we follow 4DBInfer's normalization strategy for better robustness and stability in target value distributions. The updated Relbench results are shown at Table 3

## B.3. Baseline Implementation Details

The baseline models are primarily adapted from the 4DBInfer framework, as most lacked a native implementation in RelBench. For the 4DBInfer datasets, we report the results directly from the original publication. For the RelBench datasets, while the Sage results are taken from the original report, all other baselines were re-evaluated. This process involved processing the RelBench datasets through the 4DBInfer pipeline. To ensure methodological consistency, we adhered to RelBench's hyperparameter tuning strategy and incorporated a ResNet architecture into the encoder as specified in their design.

Table 3: Comparison of RelBench Original Results and Aligned Griffin Results

| Model | rel-avito/ad-ctr | rel-f1/position | rel-hm/item-sales | rel-trial/site-success | rel-trial/study-adverse |
|---|---|---|---|---|---|
| RelBench Original Results | 0.041 | 4.022 | 0.056 | 0.400 | 44.473 |
| Aligned to Griffin Results | 0.7686 | 0.5945 | 4.440 | 0.853 | 2.199 |

This decision ensures a more reliable evaluation of model performance while maintaining consistency across tasks and datasets.

# C. Experiment and Model Details

This section provides a comprehensive overview of the experimental setup, including model updates and hyperparameter configurations. To ensure consistency and robustness, all experiments were conducted using a fixed set of model design choices and hyperparameters.

## C.1. Model Updates: Improved Cross-Attention for RDBs

In our initial model design, we applied cross-attention between task embeddings, column names, and column values to aggregate information. However, for relational databases (RDBs), we observed that in the first layer, this approach might not provide sufficient information for retrieving task-relevant columns. For example, in an RDB containing user, product, and purchase tables, predicting user-related information may be difficult without first aggregating data from the user's purchase history. This limitation was particularly evident in retrieval-related tasks, where the first-layer cross-attention often **degraded to mean aggregation**, as shown in Figure 9.

To address this issue, we modified the first-layer cross-attention to a self-attention mechanism over column names and column values. This adjustment allows the model to capture column dependencies before applying task-conditioned aggregation in later layers, improving its ability to identify relevant features. Based on experimental results, this modified approach led to better retrieval performance and overall stability, and it was set as the default configuration for all experiments.

## C.2. Hyperparameters

To ensure reproducibility, we used a fixed set of hyperparameters across all experiments. These configurations span optimization settings, model architecture, graph sampling strategies, and pretraining procedures.

For **optimization and training**, we employed the AdamW optimizer with a learning rate of 3e-4 and an L2-norm regularization of 2e-4. A batch size of 256 was used for all training runs. Early stopping was applied with a patience of 10 epochs to prevent overfitting, ensuring stable convergence. No additional learning rate scheduler or gradient clipping was used.

The **model architecture** was designed with a hidden dimension of 512, maintaining consistency between different components. The sentence embedding model was based on Nomic embeddings, truncated to 512 dimensions. The cross-attention module included 8 attention heads and a dropout rate of 0.1, allowing for effective feature extraction while preventing overfitting. SiLU was chosen as the activation function across all layers.

For **graph construction and sampling**, we adopted a 4-layer message-passing neural network (MPNN) with 2-layer uniform sampling on temporal neighbors. The fanout was set to 20 per layer to ensure a balanced trade-off between computational efficiency and capturing structural information. Additionally, reversed edges were incorporated into the sampled subgraph to improve relational modeling.

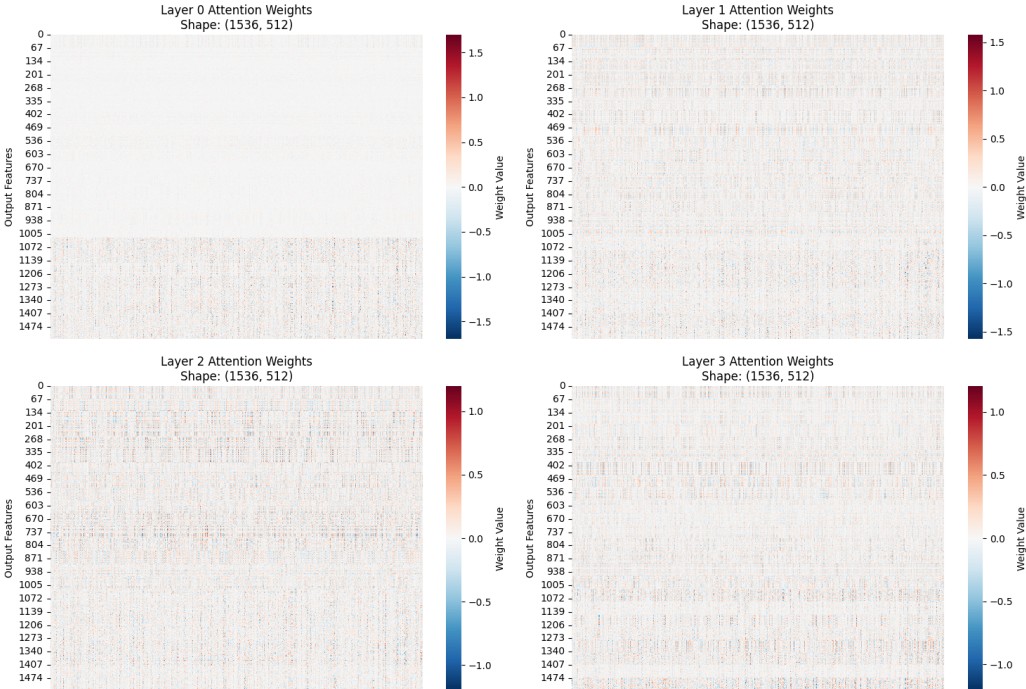

Figure 9: Cross-attention weight visualization across four layers. The heatmap shows that in the first layer, query and value activations are near zero, resulting in nearly uniform attention weights. This indicates that first-layer cross-attention effectively reduces to mean aggregation.

Regarding **pretraining and fine-tuning**, the completion pretraining phase used single-tabular datasets with early stopping signals derived from specific task performance. The same early stopping strategy was applied to supervised fine-tuning (SFT) using a combination of single-table and RDB tasks. The loss functions were cross-entropy loss for classification tasks and L2 loss for regression tasks.

The experiments were conducted on an **AWS g6.48x instance**, ensuring sufficient computational resources for large-scale graph-based training. Mixed precision (FP16) was not used, and gradient checkpointing was not applied.

These hyperparameter settings were selected to ensure a stable and scalable training process while maintaining compatibility across different relational database tasks.

## D. Extended Experiments on Joint SFT Strategies

In this section, we provide additional experiments to analyze the impact of different SFT strategies on transferability. Specifically, we investigate two key aspects: (1) The performance of different SFT strategies in transfer learning. (2) Whether domain-driven transferability conclusions remain valid across different SFT strategies.

### D.1. Experiment 1: Performance of Different SFT Strategies

To evaluate the impact of different SFT strategies, we compare five baselines:

- A no-pretrain baseline, trained directly on downstream tasks.

- A single-table-only pretrained baseline, without any RDB pretraining.

- Three joint SFT baselines, all using the same SFT domain but differing in their SFT strategies:
  - Full SFT: Standard supervised fine-tuning using all available samples.
  - Limited-Sample SFT: Fine-tuning with a restricted subset of 4096 samples.

– Mixed SFT: Joint fine-tuning with both single-tabular datasets and RDB datasets.

These baselines were evaluated on three verified transferable settings:

- Commerce-2 to Commerce-1

- Commerce-1 to Commerce-2

- Others-1 to Others-2

The results are presented in Figure 10.

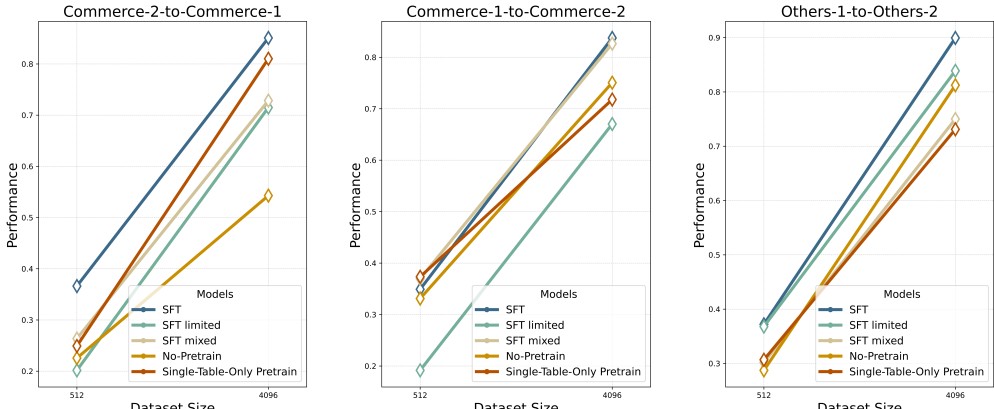

Figure 10: Performance comparison of different SFT strategies. The figure presents the results for five baselines, including no-pretraining, single-table-only pretraining, and three joint SFT strategies with varying settings. Full SFT consistently achieves the best performance across different domains, emphasizing the benefits of sufficient training.

Observing the results, we find that full SFT consistently achieves the best performance across all settings, demonstrating its superiority in transfer learning. Other baselines occasionally perform worse than the no-pretrain model, indicating their limitations in effective knowledge transfer. This further underscores the importance of sufficient training for successful adaptation.

### D.2. Experiment 2: Evaluating Domain Impact Across SFT Strategies

To determine whether domain-driven transferability conclusions remain consistent, we evaluate each SFT strategy across different SFT domains. The goal is to analyze whether transfer performance is significantly influenced by the pretraining domain. The results are presented in Figure 11 and Figure 12.

The results indicate that domain transfer effects are notably weakened under the SFT-limited strategy, suggesting that a restricted sample size hinders adaptation. In contrast, for the SFT-mixed strategy, domain transfer differences remain clearly visible, potentially indicating that SFT-mixed enables a more comprehensive adaptation compared to SFT-limited.

To further quantify the effect of domain transferability, we compute a critical difference ranking (Terpilowski, 2019) to measure average rank improvement across different settings. This serves as a straightforward yet effective method to analyze domain impact on transferability. The results are shown in Figure 13.

In general, we observe that similarity and diversity play a crucial role in transferability to "commerce" domains, reinforcing our original hypothesis. For the "others" domain, the trend remains similar but with a weakened effect. We hypothesize that incomplete SFT adaptation prevents full alignment with the pretraining domain, leading to a weaker yet general transfer effect.

These results confirm that while different SFT strategies can influence absolute transfer performance, the underlying domain-driven transferability trends remain robust.

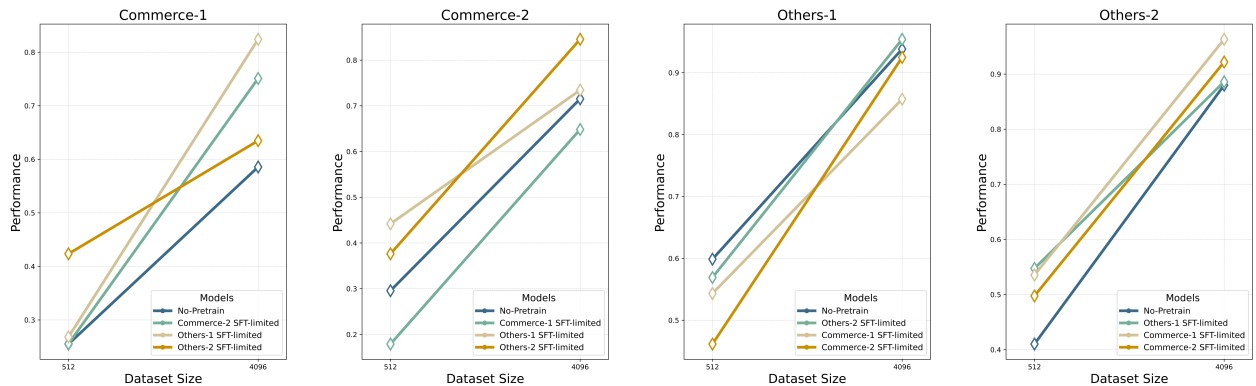

Figure 11: Impact of domain transferability under limited-sample SFT. The figure compares transfer performance across different SFT domains when fine-tuning is restricted to 4096 samples.

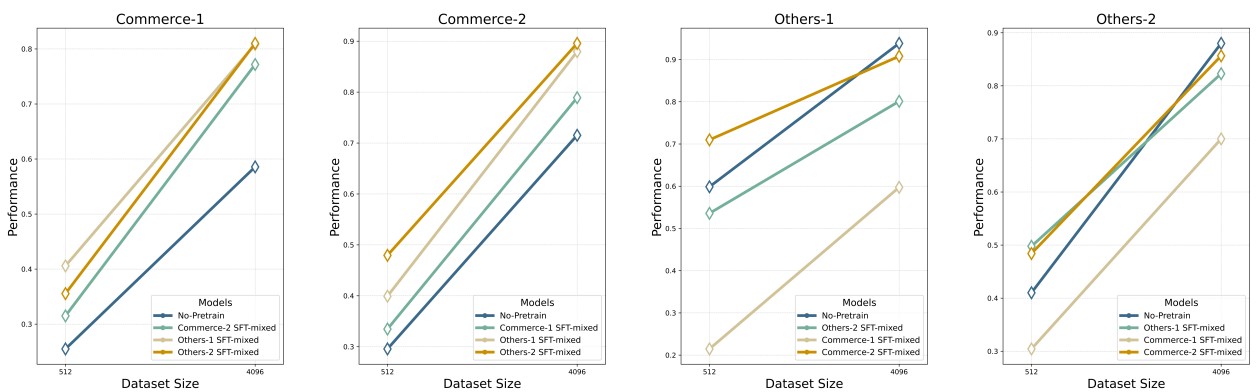

Figure 12: Impact of domain transferability under mixed SFT with single-tabular datasets. The figure evaluates whether incorporating single-tabular tasks during SFT affects transferability trends.

## E. Comparison with TabPFNv2 + DFS

This section presents a comparison between TabPFN v2 with DFS and Griffin, as shown in Figure 14. Although DFS can require several hours of preprocessing, as reported in 4DBInfer, we include it for comparison because TabPFN v2 is a strong single-table foundation model, particularly effective in few-shot settings. The results suggest that Griffin performs better on the Commerce-2 and Others-2 tasks, while TabPFN v2 shows superior results on Commerce-1 and Others-1 tasks.

## F. Raw Results

This section presents the raw experimental results for figures.

Table 4 correspond to the Figure 2. Table 5 corresponds to Figure 3. Table 6 7 8 9 correspond to Figure 4. Table 10 11 12 13 correspond to Figure 11. Table 14 15 16 17 correspond to Figure 12. Table 18 19 20 21 correspond to Figure 14.

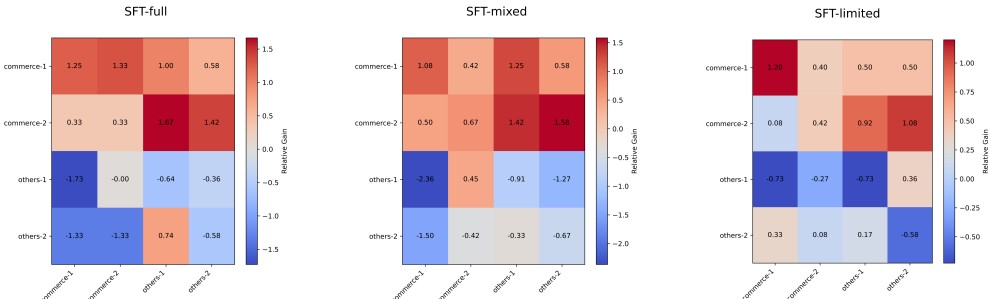

Figure 13: Critical difference ranking of domain transferability across different SFT strategies. The three subfigures represent heatmaps for Full SFT, Mixed SFT, and Limited-Sample SFT. Each cell denotes the relative gain in transfer performance from the row's domain to the column's domain, compared to the no-pretrain baseline. The diagonal cells do not represent SFT on the target domain but rather pretraining using only single-tabular datasets.

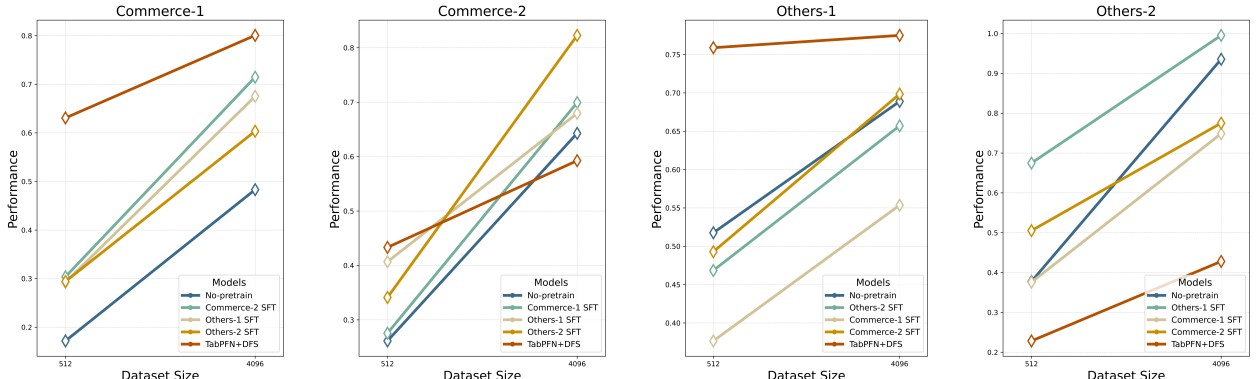

Figure 14: Evaluating Few-shot Performance: Comparison Between Griffin and TabPFNv2. This figure compares the transferability of different supervised fine-tuning strategies and TabPFNv2 in few-shot settings. Each subfigure shows the performance of five models: a no-pretraining baseline, three Griffin variants pretrained on single-table data with SFT applied to different domains, and TabPFNv2. For datasets with more than 10 classes, TabPFNv2 is not applicable; therefore, such tasks are excluded from the comparison.

Table 4: Raw Results of Griffin and All the Baselines

| Dataset/Task Metric | Overall Score Rank ↓ | Seznam/charge Accuracy ↑ | Seznam/prepay Accuracy ↑ | Airbnb/destination ROCAUC ↑ | Amazon/churn ROCAUC ↑ | Diginetica/ctr ROCAUC ↑ | Outbrain/ctr ROCAUC ↑ | Rel-avito/user-clicks ROCAUC ↑ | Rel-avito/user-visits ROCAUC ↑ |
|---|---|---|---|---|---|---|---|---|---|
| Sage | 4.792 | 0.7917 | 0.8768 | 0.8424 | 0.7358 | 0.7273 | 0.6239 | 0.6590 | 0.6620 |
| Gat | 5.208 | 0.8053 | 0.8954 | 0.8428 | 0.7410 | 0.6741 | 0.6146 | 0.6638 | 0.6417 |
| Pna | 5.333 | 0.8000 | 0.8924 | 0.8464 | 0.7645 | 0.7011 | 0.6249 | 0.6378 | 0.6267 |
| Hgt | 5.333 | 0.7965 | 0.8805 | 0.8252 | 0.7551 | 0.6733 | 0.6260 | 0.5556 | 0.6214 |
| DFS+MLP | 6.833 | 0.7554 | 0.8248 | 0.7643 | 0.6815 | 0.6944 | 0.5456 | 0.5839 | 0.6521 |
| DFS+Deepfm | 7.625 | 0.7016 | 0.8092 | 0.8007 | 0.6667 | 0.7341 | 0.5289 | 0.5912 | 0.5844 |
| DFS+Fttransformer | 5.917 | 0.7473 | 0.8162 | 0.7863 | 0.6765 | 0.7412 | 0.5360 | 0.6247 | 0.6576 |
| DFS+XGB | 7.083 | 0.7600 | 0.8453 | 0.7561 | 0.6922 | 0.7219 | 0.5421 | 0.6028 | 0.6568 |
| Griffin-unpretrained | 3.708 | 0.7998 | 0.8941 | 0.8615 | 0.7307 | 0.7157 | 0.6246 | 0.6639 | 0.6261 |
| Griffin-pretrained | 3.042 | 0.8133 | 0.9058 | 0.8681 | 0.7417 | 0.7181 | 0.6253 | 0.6330 | 0.6468 |

| Model Metric | Rel-f1/DNF ROCAUC ↑ | Rel-f1/top3 ROCAUC ↑ | Rel-hm/user-churn ROCAUC ↑ | Rel-trial/study-outcome ROCAUC ↑ | Retailrocket/cvr ROCAUC ↑ | Stackexchange/churn ROCAUC ↑ | Stackexchange/upvote ROCAUC ↑ | Virus/wnv ROCAUC ↑ | Amazon/rating RMSE ↓ |
|---|---|---|---|---|---|---|---|---|---|
| Sage | 0.7262 | 0.7554 | 0.6988 | 0.6860 | 0.8470 | 0.8558 | 0.8861 | 0.6610 | 0.9639 |
| Gat | 0.7299 | 0.7945 | 0.6788 | 0.6526 | 0.8284 | 0.8645 | 0.8853 | 0.6498 | 0.9563 |
| Pna | 0.7258 | 0.7213 | 0.5898 | 0.6558 | 0.8366 | 0.8664 | 0.8896 | 0.6684 | 0.9615 |
| Hgt | 0.7307 | 0.7226 | 0.5538 | 0.6619 | 0.8495 | 0.8670 | 0.8817 | 0.7119 | 0.9636 |
| DFS+MLP | 0.7134 | 0.7796 | 0.6802 | 0.6518 | 0.8181 | 0.8326 | 0.8783 | 0.6772 | 0.9847 |
| DFS+Deepfm | 0.6705 | 0.8133 | 0.6801 | 0.6210 | 0.8182 | 0.8212 | 0.8821 | 0.6315 | 0.9946 |
| DFS+Fttransformer | 0.7302 | 0.8250 | 0.6800 | 0.6578 | 0.8034 | 0.8376 | 0.8749 | 0.6660 | 0.9888 |
| DFS+XGB | 0.7158 | 0.8022 | 0.6786 | 0.6521 | 0.7906 | 0.8251 | 0.8675 | 0.7142 | 0.9972 |
| Griffin-unpretrained | 0.7052 | 0.7855 | 0.6847 | 0.6722 | 0.9512 | 0.8457 | 0.8956 | 0.6680 | 0.6117 |
| Griffin-pretrained | 0.7091 | 0.7795 | 0.6804 | 0.6908 | 0.9643 | 0.8435 | 0.8962 | 0.6978 | 0.5994 |

| Model Metric | Rel-avito/ad-ctr MAE ↓ | Rel-f1/position MAE ↓ | Rel-hm/item-sales MAE ↓ | Rel-trial/site-success MAE ↓ | Rel-trial/study-adverse MAE ↓ | Tel/severity Logloss ↓ | Talk/demo-pred Logloss ↓ |
|---|---|---|---|---|---|---|---|
| Sage | 1.3565 | 0.6276 | 1.1466 | 0.9191 | 1.5251 | 0.6151 | 2.3820 |
| Gat | 1.3695 | 0.6417 | 1.1086 | 0.8931 | 1.4098 | 0.6191 | 2.3880 |
| Pna | 0.8944 | 0.6084 | 1.4788 | 0.9572 | 1.4736 | 0.7015 | 2.3840 |
| Hgt | 0.6686 | 0.6212 | 1.5845 | 0.9180 | 1.5000 | 0.6126 | 2.3830 |
| DFS+MLP | 0.6610 | 0.6146 | 1.2518 | 0.9462 | 2.1104 | 0.6185 | 2.3840 |
| DFS+Deepfm | 1.1784 | 2.5796 | 1.3540 | 0.9753 | 2.0998 | 0.6118 | 2.3810 |
| DFS+Fttransformer | 0.6339 | 0.5934 | 0.9256 | 0.9793 | 1.6947 | 0.6460 | 2.3830 |
| DFS+XGB | 0.6743 | 0.6130 | 1.2957 | 0.9514 | 2.1859 | 0.6663 | 2.3890 |
| Griffin-unpretrained | 0.6593 | 0.5586 | 0.8879 | 0.7926 | 1.1700 | 0.5684 | 2.3930 |
| Griffin-pretrained | 0.6586 | 0.5694 | 0.8962 | 0.7945 | 1.2148 | 0.5518 | 2.3890 |

Table 5: Raw Results of Figure 3 on Ablation Study

| Model | Avg. Rank | Diginetica/ctr | Outbrain/ctr | Rel-f1/DNF | Rel-f1/Top3 | Rel-trial/study-outcome | Virus/wnv | Rel-avito/ad-ctr | Rel-f1/Position |
|---|---|---|---|---|---|---|---|---|---|
| Griffin | 1.4 | 0.7157 | 0.6246 | 0.7052 | 0.7855 | 0.6722 | 0.6680 | -0.6593 | -0.5586 |
| Griffin-avg-attention | 2.7 | 0.6917 | 0.6225 | 0.6936 | 0.7255 | 0.6797 | 0.5844 | -0.7600 | -0.6677 |
| Griffin-mean-GNN | 1.9 | 0.7077 | 0.6269 | 0.6950 | 0.7521 | 0.6857 | 0.6648 | -0.6701 | -0.5645 |

| Model | Rel-trial/study-adverse | Tel/severity |
|---|---|---|
| Griffin | -1.1700 | -0.5684 |
| Griffin-avg-attention | -1.3380 | -0.5449 |
| Griffin-mean-GNN | -1.2322 | -0.5694 |

Table 6: Raw Results of Figure 4 on Commerce-1 Transfer

| Dataset/Task | Diginetica/ctr | | Rel-hm/item-sales | | Rel-hm/user-churn | | Retailrocket/cvr | | Seznam/charge | | Seznam/prepay | |
|---|---|---|---|---|---|---|---|---|---|---|---|---|
| Size | 512 | 4096 | 512 | 4096 | 512 | 4096 | 512 | 4096 | 512 | 4096 | 512 | 4096 |
| No-pretrain | 0.5001 | 0.5044 | -1.2976 | -1.5236 | 0.5383 | 0.5592 | 0.7693 | 0.8002 | 0.4250 | 0.7260 | 0.5652 | 0.8180 |
| Commerce-2 SFT | 0.5213 | 0.5904 | -1.6385 | -1.4594 | 0.5677 | 0.6039 | 0.8452 | 0.9576 | 0.5662 | 0.7070 | 0.6456 | 0.7816 |
| Others-1 SFT | 0.5294 | 0.5662 | -1.8025 | -1.5055 | 0.5552 | 0.5885 | 0.8231 | 0.9446 | 0.6284 | 0.7197 | 0.7025 | 0.8030 |
| Others-2 SFT | 0.5503 | 0.5480 | -1.6472 | -1.6349 | 0.5340 | 0.5693 | 0.8006 | 0.9618 | 0.5902 | 0.7098 | 0.7060 | 0.7969 |

Table 7: Raw Results of Figure 4 on Commerce-2 Transfer

| Dataset/Task | Amazon/churn | | Amazon/rating | | Outbrain/ctr | | Rel-avito/ad-ctr | | Rel-avito/user-clicks | | Rel-avito/user-visits | |
|---|---|---|---|---|---|---|---|---|---|---|---|---|
| Size | 512 | 4096 | 512 | 4096 | 512 | 4096 | 512 | 4096 | 512 | 4096 | 512 | 4096 |
| No-pretrain | 0.5977 | 0.6580 | -0.7663 | -0.7007 | 0.4951 | 0.4990 | -0.7430 | -0.6558 | 0.5678 | 0.6187 | 0.6085 | 0.6108 |
| Commerce-1 SFT | 0.6396 | 0.6859 | -0.7740 | -0.6809 | 0.5102 | 0.5892 | -0.7267 | -0.6938 | 0.5512 | 0.5919 | 0.5779 | 0.6111 |
| Others-1 SFT | 0.5723 | 0.6675 | -0.7428 | -0.6754 | 0.5208 | 0.5096 | -0.7134 | -0.6550 | 0.6078 | 0.6056 | 0.6129 | 0.6198 |
| Others-2 SFT | 0.6231 | 0.6645 | -0.7513 | -0.6797 | 0.5245 | 0.6163 | -0.7098 | -0.6518 | 0.5481 | 0.5951 | 0.5944 | 0.6275 |

Table 8: Raw Results of Figure 4 on Others-1 Transfer

| Dataset/Task | Rel-f1/DNF | | Rel-f1/position | | Rel-f1/top3 | | Stackexchange/churn | | Stackexchange/upvote | | Virus/wnv | |
|---|---|---|---|---|---|---|---|---|---|---|---|---|
| Size | 512 | 4096 | 512 | 4096 | 512 | 4096 | 512 | 4096 | 512 | 4096 | 512 | 4096 |
| No-pretrain | 0.6558 | 0.7176 | -0.6152 | -0.5746 | 0.7676 | N/A | 0.7256 | 0.7951 | 0.8433 | 0.8772 | 0.6099 | 0.6652 |
| Others-2 SFT | 0.7043 | 0.7248 | -0.7056 | -0.5921 | 0.7900 | N/A | 0.7188 | 0.7928 | 0.8571 | 0.8731 | 0.5706 | 0.6567 |
| Commerce-1 SFT | 0.6823 | 0.7233 | -0.6221 | -0.6043 | 0.6831 | N/A | 0.6602 | 0.7501 | 0.8107 | 0.8597 | 0.6101 | 0.6461 |
| Commerce-2 SFT | 0.6073 | 0.7347 | -0.6255 | -0.5931 | 0.7679 | N/A | 0.7600 | 0.8164 | 0.8544 | 0.8745 | 0.5946 | 0.6591 |

Table 9: Raw Results of Figure 4 on Others-2 Transfer

| Dataset/Task | Airbnb/destination | | Rel-trial/site-success | | Rel-trial/study-adverse | | Rel-trial/study-outcome | | Talk/demo-pred | | Tel/severity | |
|---|---|---|---|---|---|---|---|---|---|---|---|---|
| Size | 512 | 4096 | 512 | 4096 | 512 | 4096 | 512 | 4096 | 512 | 4096 | 512 | 4096 |
| No-pretrain | 0.8562 | 0.8564 | -0.9327 | -0.9062 | -2.5758 | -1.3676 | 0.6283 | 0.6656 | -2.4363 | -2.4245 | -0.7910 | -0.5998 |
| Others-1 SFT | 0.8470 | 0.8561 | -0.9142 | -0.8952 | -1.5832 | -1.3246 | 0.6436 | 0.6781 | -2.4407 | -2.4177 | -0.7716 | -0.6052 |
| Commerce-1 SFT | 0.8457 | 0.8543 | -0.9231 | -0.9345 | -1.9609 | -1.5188 | 0.5583 | 0.6493 | -2.4377 | -2.4177 | -0.7965 | -0.6523 |
| Commerce-2 SFT | 0.8007 | 0.8516 | -0.9363 | -0.9496 | -1.6756 | -1.3703 | 0.6001 | 0.6567 | -2.4394 | -2.4164 | -0.7721 | -0.6266 |

Table 10: Raw Results of Figure 11 on Commerce-1 Transfer

| Dataset/Task | Diginetica/ctr | | Rel-hm/item-sales | | Rel-hm/user-churn | | Retailrocket/cvr | | Seznam/charge | | Seznam/prepay | |
|---|---|---|---|---|---|---|---|---|---|---|---|---|
| Size | 512 | 4096 | 512 | 4096 | 512 | 4096 | 512 | 4096 | 512 | 4096 | 512 | 4096 |
| No-pretrain | 0.5001 | 0.5044 | -1.2976 | -1.5236 | 0.5383 | 0.5592 | 0.7693 | 0.8002 | 0.4250 | 0.7260 | 0.5652 | 0.8180 |
| Commerce-2 SFT | 0.4655 | 0.4969 | -1.6560 | -1.4736 | 0.5376 | 0.5728 | 0.8368 | 0.9577 | 0.5119 | 0.7214 | 0.6478 | 0.8091 |
| Others-1 SFT | 0.5266 | 0.5744 | -1.7979 | -1.4878 | 0.5465 | 0.5704 | 0.8512 | 0.9601 | 0.5499 | 0.7237 | 0.5557 | 0.8075 |
| Others-2 SFT | 0.5425 | 0.5465 | -1.5732 | -1.6291 | 0.5583 | 0.5719 | 0.8848 | 0.9215 | 0.5641 | 0.7195 | 0.5563 | 0.6780 |

Table 11: Raw Results of Figure 11 on Commerce-2 Tasks

| Dataset/Task | Amazon/churn | | Amazon/rating | | Outbrain/ctr | | Rel-avito/ad-ctr | | Rel-avito/user-clicks | | Rel-avito/user-visits | |
|---|---|---|---|---|---|---|---|---|---|---|---|---|
| Size | 512 | 4096 | 512 | 4096 | 512 | 4096 | 512 | 4096 | 512 | 4096 | 512 | 4096 |
| No-pretrain | 0.5977 | 0.6580 | -0.7663 | -0.7007 | 0.4951 | 0.4990 | -0.7430 | -0.6558 | 0.5678 | 0.6187 | 0.6085 | 0.6108 |
| Commerce-1 SFT | 0.5970 | 0.6382 | -0.7480 | -0.6941 | 0.5126 | 0.5080 | -0.7115 | -0.6503 | 0.5028 | 0.5741 | 0.5275 | 0.6050 |
| Others-1 SFT | 0.6118 | 0.6673 | -0.7541 | -0.6857 | 0.4930 | 0.5008 | -0.7097 | -0.6522 | 0.6047 | 0.6009 | 0.6090 | 0.6099 |
| Others-2 SFT | 0.6147 | 0.6624 | -0.7540 | -0.6783 | 0.5085 | 0.5873 | -0.7260 | -0.6579 | 0.5568 | 0.6062 | 0.6112 | 0.6081 |

Table 12: Raw Results of Figure 11 on Others-1 Tasks

| Dataset/Task | Rel-f1/DNF | | Rel-f1/position | | Rel-f1/top3 | | Stackexchange/churn | | Stackexchange/upvote | | Virus/wnv | |
|---|---|---|---|---|---|---|---|---|---|---|---|---|
| Size | 512 | 4096 | 512 | 4096 | 512 | 4096 | 512 | 4096 | 512 | 4096 | 512 | 4096 |
| No-pretrain | 0.6558 | 0.7176 | -0.6152 | -0.5746 | 0.7676 | N/A | 0.7256 | 0.7951 | 0.8433 | 0.8772 | 0.6099 | 0.6652 |
| Others-2 SFT | 0.7098 | 0.7275 | -0.6680 | -0.5739 | 0.7710 | N/A | 0.6592 | 0.7898 | 0.8530 | 0.8761 | 0.6148 | 0.6737 |
| Commerce-1 SFT | 0.6752 | 0.7226 | -0.6089 | -0.5813 | 0.7375 | N/A | 0.6477 | 0.7337 | 0.8406 | 0.8770 | 0.6397 | 0.6603 |
| Commerce-2 SFT | 0.5808 | 0.7137 | -0.6433 | -0.5752 | 0.7536 | N/A | 0.6526 | 0.7839 | 0.8605 | 0.8779 | 0.6265 | 0.6676 |

Table 13: Raw Results of Figure 11 on Others-2 Tasks

| Dataset/Task | Airbnb/destination | | Rel-trial/site-success | | Rel-trial/study-adverse | | Rel-trial/study-outcome | | Talk/demo-pred | | Tel/severity | |
|---|---|---|---|---|---|---|---|---|---|---|---|---|
| Size | 512 | 4096 | 512 | 4096 | 512 | 4096 | 512 | 4096 | 512 | 4096 | 512 | 4096 |
| No-pretrain | 0.8562 | 0.8564 | -0.9327 | -0.9062 | -2.5758 | -1.3676 | 0.6283 | 0.6656 | -2.4363 | -2.4245 | -0.7910 | -0.5998 |
| Others-1 SFT | 0.8489 | 0.8550 | -0.9156 | -0.9135 | -1.6351 | -1.3448 | 0.6163 | 0.6730 | -2.4309 | -2.4160 | -0.8007 | -0.6114 |
| Commerce-1 SFT | 0.8480 | 0.8546 | -0.9220 | -0.8820 | -1.5550 | -1.3571 | 0.6020 | 0.6639 | -2.4318 | -2.4142 | -0.7800 | -0.6001 |
| Commerce-2 SFT | 0.8427 | 0.8534 | -0.9332 | -0.8927 | -1.5329 | -1.3321 | 0.6338 | 0.6834 | -2.4432 | -2.4222 | -0.7875 | -0.6208 |

Table 14: Raw Results of Figure 12 on Commerce-1 Tasks

| Dataset/Task | Diginetica/ctr | | Rel-hm/item-sales | | Rel-hm/user-churn | | Retailrocket/cvr | | Seznam/charge | | Seznam/prepay | |
|---|---|---|---|---|---|---|---|---|---|---|---|---|
| Size | 512 | 4096 | 512 | 4096 | 512 | 4096 | 512 | 4096 | 512 | 4096 | 512 | 4096 |
| No-pretrain | 0.5001 | 0.5044 | -1.2976 | -1.5236 | 0.5383 | 0.5592 | 0.7693 | 0.8002 | 0.4250 | 0.7260 | 0.5652 | 0.8180 |
| Commerce-2 SFT | 0.4786 | 0.5145 | -1.6061 | -1.5209 | 0.5492 | 0.5835 | 0.8302 | 0.9675 | 0.5689 | 0.7089 | 0.6101 | 0.7952 |
| Others-1 SFT | 0.4808 | 0.5316 | -1.5613 | -1.4722 | 0.5525 | 0.5834 | 0.8130 | 0.9612 | 0.6408 | 0.7231 | 0.6749 | 0.7958 |
| Others-2 SFT | 0.4500 | 0.5539 | -1.4179 | -1.5491 | 0.5528 | 0.5938 | 0.8009 | 0.9630 | 0.6092 | 0.7026 | 0.6139 | 0.7776 |

Table 15: Raw Results of Figure 12 on Commerce-2 Tasks

| Dataset/Task | Amazon/churn | | Amazon/rating | | Outbrain/ctr | | Rel-avito/ad-ctr | | Rel-avito/user-clicks | | Rel-avito/user-visits | |
|---|---|---|---|---|---|---|---|---|---|---|---|---|
| Size | 512 | 4096 | 512 | 4096 | 512 | 4096 | 512 | 4096 | 512 | 4096 | 512 | 4096 |
| No-pretrain | 0.5977 | 0.6580 | -0.7663 | -0.7007 | 0.4951 | 0.4990 | -0.7430 | -0.6558 | 0.5678 | 0.6187 | 0.6085 | 0.6108 |
| Commerce-1 SFT | 0.6220 | 0.6730 | -0.7678 | -0.6861 | 0.4966 | 0.5659 | -0.7428 | -0.6630 | 0.5812 | 0.5794 | 0.6000 | 0.6162 |
| Others-1 SFT | 0.6241 | 0.6634 | -0.7583 | -0.6824 | 0.5208 | 0.6134 | -0.7418 | -0.6723 | 0.5858 | 0.6229 | 0.6046 | 0.6130 |
| Others-2 SFT | 0.6278 | 0.6592 | -0.7864 | -0.7019 | 0.5553 | 0.6206 | -0.7070 | -0.6614 | 0.5724 | 0.6262 | 0.6212 | 0.6238 |

Table 16: Raw Results of Figure 12 on Others-1 Tasks

| Dataset/Task | Rel-f1/DNF | | Rel-f1/position | | Rel-f1/top3 | | Stackexchange/churn | | Stackexchange/upvote | | Virus/wnv | |
|---|---|---|---|---|---|---|---|---|---|---|---|---|
| Size | 512 | 4096 | 512 | 4096 | 512 | 4096 | 512 | 4096 | 512 | 4096 | 512 | 4096 |
| No-pretrain | 0.6558 | 0.7176 | -0.6152 | -0.5746 | 0.7676 | N/A | 0.7256 | 0.7951 | 0.8433 | 0.8772 | 0.6099 | 0.6652 |
| Others-2 SFT | 0.6820 | 0.7123 | -0.6000 | -0.5955 | 0.7521 | N/A | 0.6448 | 0.7691 | 0.8414 | 0.8670 | 0.6041 | 0.6335 |
| Commerce-1 SFT | 0.6492 | 0.7131 | -0.7286 | -0.6978 | 0.6441 | N/A | 0.7188 | 0.7552 | 0.8384 | 0.8599 | 0.5259 | 0.6075 |
| Commerce-2 SFT | 0.7123 | 0.7229 | -0.6226 | -0.5982 | 0.7698 | N/A | 0.7429 | 0.8111 | 0.8491 | 0.8726 | 0.6315 | 0.6564 |

Table 17: Raw Results of Figure 12 on Others-2 Tasks

| Dataset/Task | Airbnb/destination | | Rel-trial/site-success | | Rel-trial/study-adverse | | Rel-trial/study-outcome | | Talk/demo-pred | | Tel/severity | |
|---|---|---|---|---|---|---|---|---|---|---|---|---|
| Size | 512 | 4096 | 512 | 4096 | 512 | 4096 | 512 | 4096 | 512 | 4096 | 512 | 4096 |
| No-pretrain | 0.8562 | 0.8564 | -0.9327 | -0.9062 | -2.5758 | -1.3676 | 0.6283 | 0.6656 | -2.4363 | -2.4245 | -0.7910 | -0.5998 |
| Others-1 SFT | 0.8512 | 0.8548 | -0.9245 | -0.9246 | -1.6203 | -1.3474 | 0.5977 | 0.6637 | -2.4396 | -2.4225 | -0.7844 | -0.6151 |
| Commerce-1 SFT | 0.8388 | 0.8541 | -0.9272 | -0.9362 | -1.8690 | -1.4631 | 0.5729 | 0.6460 | -2.4710 | -2.4219 | -0.7749 | -0.6815 |
| Commerce-2 SFT | 0.8477 | 0.8518 | -0.9357 | -0.9177 | -1.6238 | -1.3327 | 0.6143 | 0.6599 | -2.4373 | -2.4139 | -0.7896 | -0.6112 |

Table 18: Raw Results of Figure 14 on Comparison with TabPFN on Commerce-1 Transfer

| Dataset/Task | Diginetica/ctr | | Rel-hm/item-sales | | Rel-hm/user-churn | | Retailrocket/cvr | | Seznam/charge | | Seznam/prepay | |
|---|---|---|---|---|---|---|---|---|---|---|---|---|
| Size | 512 | 4096 | 512 | 4096 | 512 | 4096 | 512 | 4096 | 512 | 4096 | 512 | 4096 |
| No-pretrain | 0.5001 | 0.5044 | -1.2976 | -1.5236 | 0.5383 | 0.5592 | 0.7693 | 0.8002 | 0.4250 | 0.7260 | 0.5652 | 0.8180 |
| Commerce-2 SFT | 0.5213 | 0.5904 | -1.6385 | -1.4594 | 0.5677 | 0.6039 | 0.8452 | 0.9576 | 0.5662 | 0.7070 | 0.6456 | 0.7816 |
| Others-1 SFT | 0.5294 | 0.5662 | -1.8025 | -1.5055 | 0.5552 | 0.5885 | 0.8231 | 0.9446 | 0.6284 | 0.7197 | 0.7025 | 0.8030 |
| Others-2 SFT | 0.5503 | 0.5480 | -1.6472 | -1.6349 | 0.5340 | 0.5693 | 0.8006 | 0.9618 | 0.5902 | 0.7098 | 0.7060 | 0.7969 |
| TabPFN+DFS | 0.6696 | 0.7588 | -1.5817 | -1.3855 | 0.6647 | 0.6746 | 0.7769 | 0.7928 | 0.7098 | 0.7289 | 0.7639 | 0.7817 |

Table 19: Raw Results of Figure 14 on Comparison with TabPFN on Commerce-2 Transfer

| Dataset/Task | Amazon/churn | | Amazon/rating | | Outbrain/ctr | | Rel-avito/ad-ctr | | Rel-avito/user-clicks | | Rel-avito/user-visits | |
|---|---|---|---|---|---|---|---|---|---|---|---|---|
| Size | 512 | 4096 | 512 | 4096 | 512 | 4096 | 512 | 4096 | 512 | 4096 | 512 | 4096 |
| No-pretrain | 0.5977 | 0.6580 | -0.7663 | -0.7007 | 0.4951 | 0.4990 | -0.7430 | -0.6558 | 0.5678 | 0.6187 | 0.6085 | 0.6108 |
| Commerce-1 SFT | 0.6396 | 0.6859 | -0.7740 | -0.6809 | 0.5102 | 0.5892 | -0.7267 | -0.6938 | 0.5512 | 0.5919 | 0.5779 | 0.6111 |
| Others-1 SFT | 0.5723 | 0.6675 | -0.7428 | -0.6754 | 0.5208 | 0.5096 | -0.7134 | -0.6550 | 0.6078 | 0.6056 | 0.6129 | 0.6198 |
| Others-2 SFT | 0.6231 | 0.6645 | -0.7513 | -0.6797 | 0.5245 | 0.6163 | -0.7098 | -0.6518 | 0.5481 | 0.5951 | 0.5944 | 0.6275 |
| TabPFN+DFS | 0.6283 | 0.6351 | -1.0317 | -1.0032 | 0.5211 | 0.5382 | -0.7034 | -0.6913 | 0.6125 | 0.6395 | 0.6380 | 0.6576 |

Table 20: Raw Results of Figure 14 on Comparison with TabPFN on Others-1 Transfer

| Dataset/Task | Rel-f1/DNF | | Rel-f1/position | | Rel-f1/top3 | | Stackexchange/churn | | Stackexchange/upvote | | Virus/wnv | |
|---|---|---|---|---|---|---|---|---|---|---|---|---|
| Size | 512 | 4096 | 512 | 4096 | 512 | 4096 | 512 | 4096 | 512 | 4096 | 512 | 4096 |
| No-pretrain | 0.6558 | 0.7176 | -0.6152 | -0.5746 | 0.7676 | N/A | 0.7256 | 0.7951 | 0.8433 | 0.8772 | 0.6099 | 0.6652 |
| Others-2 SFT | 0.7043 | 0.7248 | -0.7056 | -0.5921 | 0.7900 | N/A | 0.7188 | 0.7928 | 0.8571 | 0.8731 | 0.5706 | 0.6567 |
| Commerce-1 SFT | 0.6823 | 0.7233 | -0.6221 | -0.6043 | 0.6831 | N/A | 0.6602 | 0.7501 | 0.8107 | 0.8597 | 0.6101 | 0.6461 |
| Commerce-2 SFT | 0.6073 | 0.7347 | -0.6255 | -0.5931 | 0.7679 | N/A | 0.7600 | 0.8164 | 0.8544 | 0.8745 | 0.5946 | 0.6591 |
| TabPFN+DFS | 0.7120 | 0.7346 | -0.6587 | -0.5962 | 0.8003 | N/A | 0.7886 | 0.8212 | 0.8562 | 0.8649 | 0.7666 | 0.7905 |

Table 21: Raw Results of Figure 14 on Comparison with TabPFN on Others-2 Transfer

| Dataset/Task | Airbnb/destination | | Rel-trial/site-success | | Rel-trial/study-adverse | | Rel-trial/study-outcome | | Talk/demo-pred | | Tel/severity | |
|---|---|---|---|---|---|---|---|---|---|---|---|---|
| Size | 512 | 4096 | 512 | 4096 | 512 | 4096 | 512 | 4096 | 512 | 4096 | 512 | 4096 |
| No-pretrain | 0.8562 | 0.8564 | -0.9327 | -0.9062 | -2.5758 | -1.3676 | 0.6283 | 0.6656 | -2.4363 | -2.4245 | -0.7910 | -0.5998 |
| Others-1 SFT | 0.8470 | 0.8561 | -0.9142 | -0.8952 | -1.5832 | -1.3246 | 0.6436 | 0.6781 | -2.4407 | -2.4177 | -0.7716 | -0.6052 |
| Commerce-1 SFT | 0.8457 | 0.8543 | -0.9231 | -0.9345 | -1.9609 | -1.5188 | 0.5583 | 0.6493 | -2.4377 | -2.4177 | -0.7965 | -0.6523 |
| Commerce-2 SFT | 0.8007 | 0.8516 | -0.9363 | -0.9496 | -1.6756 | -1.3703 | 0.6001 | 0.6567 | -2.4394 | -2.4164 | -0.7721 | -0.6266 |
| TabPFN+DFS | N/A | N/A | -0.9687 | -0.9873 | -1.9763 | -1.6555 | 0.5861 | 0.6602 | N/A | N/A | -0.8865 | -0.8505 |

