# OpenReview forum: "Griffin: Towards a Graph-Centric Relational Database Foundation Model"
_ICML.cc/2025/Conference — ICML 2025 poster_

### Official Review · Reviewer_95m6 · 2025-02-28

**Overall Recommendation:** 4

**Summary:**

This paper proposes Griffin, a unified model for Relational DataBase(RDB). Griffin has a classification prediction head and a regression prediction head to handle the classification and regression tasks, correspondingly. The classification head can handle arbitrary classes by comparing the inner product between the generated node representation and the text embeddings of the target categories. Components like cross-attention modules and unified task formats have been proposed to make this one model suitable for different scenarios. Griffin follows a pretrain-finetune paradigm. It first pretrains on a set of single tables and RDBs. On a specific task, it is then further finetuned on the provided data. Experiments show that Griffin, especially the pretrained one, performs better than the GraphSAGE baseline.

**Claims And Evidence:**

NA

**Essential References Not Discussed:**

Past work:

Relational Deep Learning: Graph Representation Learning on Relational Databases. Matthias Fey, Weihua Hu, Kexin Huang, Jan Eric Lenssen, Rishabh Ranjan, Joshua Robinson, Rex Ying, Jiaxuan You, Jure Leskovec

Contemporary work:

RelGNN: Composite Message Passing for Relational Deep Learning. Tianlang Chen, Charilaos Kanatsoulis, Jure Leskovec

ContextGNN: Beyond Two-Tower Recommendation Systems. Yiwen Yuan, Zecheng Zhang, Xinwei He, Akihiro Nitta, Weihua Hu, Dong Wang, Manan Shah, Shenyang Huang, Blaž Stojanovič, Alan Krumholz, Jan Eric Lenssen, Jure Leskovec, Matthias Fey

**Experimental Designs Or Analyses:**

- For the single-table tasks, more baselines should be included for a comprehensive comparison, like XgBoost, LightGBM and TabPFN[1].

[1] TabPFN: A Transformer That Solves Small Tabular Classification Problems in a Second. Noah Hollmann, Samuel Müller, Katharina Eggensperger, Frank Hutter

**Methods And Evaluation Criteria:**

- The proposed method is properly evaluated on the benchmark datasets.
- The significance of the proposed Griffin needs to be further validated. Standard deviation needs to be included to verify the significance.

**Other Comments Or Suggestions:**

NA

**Other Strengths And Weaknesses:**

Strength:

- The task over RDB is a practical problem and has a great impact on the real-world applications.

- The writing of the paper is clear and easy to understand.

- The tailored design of Griffin makes it able to perform arbitrary classification tasks and regression tasks in a single model.

Weakness:

- The experiment part is not comprehensive. There is only GraphSAGE as the baseline.

- Several design choices are not well supported by evidence. For example, what is the advantage of the Cross Attention Module compared to just averaging the embedding? Similarly, during message-passing, why use Max aggregation rather than Mean (Attention) to aggregate messages from different edge types? Why the encoder/decoder for numeric features are trained seperately rather than jointly?

**Questions For Authors:**

- The encoder/decoder of the numerical features is trained alone, apart from the entire training pipeline. What is the benefit of this? There is no clear reason why it can't be trained end-to-end. To support the claim, an ablation study is needed to validate the design.

- When encoding the target node, how do you generate the embedding of the missing column? I assume there will be a default missing embedding for the missing value.

- What if there are missing values in multiple nodes that are not the target node?

**Relation To Broader Scientific Literature:**

NA

**Theoretical Claims:**

NA

---

> ### Author Rebuttal · Authors · 2025-04-01
>
> We thank Reviewer 95m6 for acknowledging the contributions of our work and for the helpful suggestions and questions. We address each concern below. The updated experiments including 7 additional baselines about main experiment and 2 additional baselines on few-shot setting is provided at https://anonymous.4open.science/r/Griffin-Rebuttal
>
> ---
>
> **Related Work**
> Thank you for the recommended references. In response to your suggestions, along with those from other reviewers, we have revised the related work section to include a broader range of relevant literature. We have added new subsections covering embedding-based RDB models and Table QA approaches, as well as additional works on RDB-specific models and tabular foundation models.
>
> To better reflect the landscape and clarify our positioning, we now organize the related work into the following categories:
> 	•	Single-table and RDB models for tabular predictive tasks
> 	•	Table QA and SQL generation methods
> 	•	Tabular and graph foundation models, including both LLM-based models and those trained from scratch
>
> ---
>
> **W1: Experimental Coverage and Baselines**
>
> Thank you for your concern regarding the experimental baselines. We have significantly extended our experiments by including the following additional baseline categories:
>
> 1.	**Other GNN-based models** for RDBs, including **GAT**, **PNA**, and **HGT**.
>
> 2.	**Widely used single-table methods**, such as **XGBoost**, **MLP**, **DeepFM**, and **Feature Transformer**, applied to RDBs via **Deep Feature Synthesis (DFS)** to flatten the schema.
>
> 3.	**Tabular foundation models**, including **TabPFN** and **TabLLM**, also used with DFS.
>
> These baselines are now incorporated into the experimental section, reinforcing its effectiveness across RDB-related tasks.
>
> ---
>
> **W2 & Q1: Design Choices and Ablation Studies**
>
> Thank you for raising questions about the justification of specific design choices. We have now addressed these through targeted ablation studies and reorganized the section to improve readability. Specifically:
>
> •	**Cross-Attention vs. Mean Pooling**: We compare our cross-attention mechanism with simple mean-pooling. The results show improvements, supporting the benefit of selective attention over equal-weighted aggregation.
>
> •	**Max vs. Mean Aggregation**: We evaluate different aggregation methods over heterogeneous relations. Max aggregation helps highlight the most informative relation types, and the ablation supports this design choice.
>
> •	**Numeric Encoder/Decoder Pretraining**: Our design choice is based on the goal of enabling transferability across tasks. To ensure that the model outputs remain in a consistent numerical space, we use a fixed numeric encoder and decoder during training. Since our primary focus is not on improving single-task performance but rather on transferability, we did not include ablation studies on single-task performance for this component. That said, we recognize the importance of this question and are planning a more comprehensive ablation study in future work—starting from re-pretraining the model to investigate the impact of this design on cross-task generalization.
>
> ---
>
> **Q2 & Q3: Handling Missing Values**
>
> We appreciate the questions on how missing values are handled, both in the target node and in neighboring nodes:
>
> •	**For the target node’s missing column (label)**: We mask it using a **zero placeholder** during pretraining.
>
> •	**For other missing values**:
>
> •	**Categorical/Text features** are replaced with a "None" token.
>
> •	**Numeric features** are replaced with the **column-wise mean**.
>
> This strategy generalizes to cases where **multiple features or nodes have missing values**, and aligns with common practices in both tabular learning and graph-based models. We have clarified this in the methodology section.

---

> > ### Comment · Reviewer_95m6 · 2025-04-02
> >
> > Thank the authors for addressing my questions. The additional results at the rebuttal stage should be included in the final revision.

---

> > > ### Author Response · Authors · 2025-04-03
> > >
> > > Thank you for acknowledging our work! We’ve updated the key experimental figures and will further refine them in the final version.

---

### Official Review · Reviewer_XrAk · 2025-03-09

**Overall Recommendation:** 3

**Summary:**

This paper proposes Griffin, a foundation model specifically designed for Relational Databases (RDBs), which leverages graph neural networks (GNNs) to unify the processing of diverse RDB tasks. Experiments demonstrate that Griffin exhibits superior or comparable performance across multiple benchmarks, especially in low-data scenarios.

**Claims And Evidence:**

Partially. The authors do not provide a clear description of  the challenges in developing the foundation models for RDBs.

**Essential References Not Discussed:**

No.

**Experimental Designs Or Analyses:**

While the experiments comprehensively evaluate Griffin's performance across diverse tasks and datasets, the study primarily focuses on validating the proposed model itself. Notably, the absence of comparative analyses with existing models or methods limits the understanding of how Griffin's capabilities measure against alternative approaches in solving RDB-related problems.

**Methods And Evaluation Criteria:**

Partially. The authors highlight that Griffin is capable of handling temporal heterogeneous graphs, a feature that distinguishes it from traditional Message-Passing Neural Networks (MPNNs), as the latter does not inherently support temporal data processing.

**Other Comments Or Suggestions:**

In Section 1, Paragraph 2, the connection between graph-based methods and RDB problems is not clearly articulated. Additionally, the visualization of main results in Figures 2 and 3 needs refinement. The current vertical labeling of the x-axis significantly hinders readability, making it challenging to interpret the data effectively. A more user-friendly design, such as horizontal x-axis labels, would greatly enhance the clarity and accessibility of these critical findings.

**Other Strengths And Weaknesses:**

Strengths:

1. The paper presents a foundation model specifically designed for RDB-related problems, effectively addressing a significant gap in the field.

2. Comprehensive experimental results demonstrate the model's robust and comparable performance across multiple RDB-related tasks, showcasing its versatility.

Weakness:

1. The paper's organization and presentation could be improved for better clarity. For example, Section 1 Paragraph 4 introduces the first challenge, while the subsequent paragraph (Paragraph 5) shifts to discussing Griffin's advantages, creating a disjointed flow that may confuse readers.

2. The paper lacks a clear justification for the necessity of developing a specialized foundation model for RDB-related tasks. It does not adequately address whether existing frameworks, such as those based on large language models, could achieve similar or better results.

3. The experimental evaluation is insufficient. The paper fails to compare Griffin's performance against the state-of-the-art (SOTA) methods for RDB-related tasks, relying instead on a single baseline (Sage) that does not represent SOTA. Additionally, the authors do not provide ablation studies or analyses to demonstrate the contribution and necessity of each component in the proposed model framework.

**Questions For Authors:**

1. The authors should address the necessity of developing a foundation model specifically for RDB-related problems. It remains unclear whether existing models or frameworks could effectively solve these tasks. To strengthen the significance of this work, the authors should provide comparative results against commonly used frameworks, as the absence of such comparisons undermines the motivation and relevance of the proposed approach.

2. The relationship between Griffin and temporal data requires further clarification. While the paper highlights Griffin's capability to handle heterogeneous RDB data, it lacks a detailed explanation of how the model processes temporal data. This omission creates ambiguity regarding the model's applicability to time-dependent RDB tasks, which is a critical aspect of real-world database systems.

**Relation To Broader Scientific Literature:**

The Griffin is built upon the Graph Foundation Models (GFMs) and the Tabular Foundation Models (TFMs) and extend them to the RDB-related problems.

**Theoretical Claims:**

N/A

---

> ### Author Rebuttal · Authors · 2025-04-01
>
> We thank Reviewer XrAk for the thoughtful and constructive feedback. We address your concerns in detail below. The updated experiments including 7 additional baselines about main experiment and 2 additional baselines on few-shot setting is provided at https://anonymous.4open.science/r/Griffin-Rebuttal
>
> ---
>
> **Claim 1: Clarifying the Challenges in Developing Foundation Models for RDBs**
>
> Thank you for pointing this out. We agree that the challenges could be stated more clearly. We now explicitly outline the key challenges in developing a foundation model for relational databases:
>
> 1.	**Data and task diversity**: RDBs contain various data types (numerical, categorical, textual) and support diverse tasks, including regression and multi-class classification. A unified model must handle all these variations under a single framework.
>
> 2.	**Lack of RDB-specific GNNs**: Existing GNN architectures are not tailored for relational databases and often fail to leverage the rich metadata (e.g., table schema, column descriptions, task semantics) that is critical in RDBs.
>
> 3.	**Missing pretraining pipeline and transferability analysis**: It is not yet clear how pretraining can benefit relational data, nor what kinds of tasks or data lead to effective transferability across RDBs.
>
> These challenges directly motivate the three components of Griffin: unified encoder/decoder design, GNN architecture enhancements, and a new pretraining pipeline. We have now added a dedicated subsection to the paper to clearly articulate these challenges and align them with our proposed solutions.
>
> ---
>
> **Experiment: Including More Baselines & W3: Ablations and Experimental Focus**
>
> Thank you for raising this point. While our primary goal is to present a unified foundation model framework for RDBs, we agree that including broader baselines and detailed ablation studies is essential for a comprehensive evaluation.
>
> To that end, we have expanded our experimental comparisons to include three additional categories of baselines:
>
> • **Three additional GNN-based baselines**: GAT, PNA, and HGT, as suggested in 4DBInfer.
>
> • **Four single-table baselines**: MLP, DeepFM, Feature Transformer, and XGBoost, along with **Deep Feature Synthesis (DFS)**—a strong feature synthesis method that converts multi-table data into a single table by computing meaningful feature combinations.
>
> • **Two single-table baselines for few-shot settings**: We include TabPFN and TabLLM as representative few-shot baselines, both of which leverage pretrained models. TabPFN is trained from scratch, while TabLLM uses LLM-based pretraining.
>
> We have also reorganized the experimental section to make the ablation studies more prominent. These ablations highlight the contributions of key architectural choices—specifically, the **cross-attention mechanism** and **hierarchical aggregation**—to the overall performance of the model.
>
> ---
>
> **W1: Organization and Presentation**
>
> Thanks for your suggestions. We have revised the introduction to better connect the discussion of challenges with our method overview.
>
> ---
>
> **W2 & Q1: Justification for RDB-Specific Foundation Models**
>
> We appreciate this important question. While single-table models can be combined with DFS to simulate multi-table input, they inherently lack the ability to capture the complex relational structure that is central to RDBs. This limitation has been highlighted in prior benchmarks such as 4DBInfer, where structure-aware models (e.g., GNN-based) consistently outperform single-table methods with DFS.
>
> To further support this, we have added experimental comparisons with strong single-table baselines, including XGBoost, DeepFM, TabPFN, and TabLLM (all paired with DFS). The results reinforce the need for RDB-specific model designs like Griffin.
>
> ---
>
> **Comments and Q2: Handling Temporal Data**
>
> Thank you for this valuable observation. Our approach to handling temporal information follows the setup used in 4DBInfer, where subgraphs are constructed such that all neighboring nodes have timestamps earlier than the target node. This ensures that the model only uses past information, aligning with the temporal nature of real-world relational data.
>
> To improve clarity, we have **expanded the explanation** in the preliminaries and **reorganized the method section** to better highlight how temporal constraints are applied during subgraph construction.
>
> Additionally, we have revised the figures and included more baselines to improve presentation.
>
> We appreciate your feedback on presentation—it has significantly improved the clarity and accessibility of our results.

---

### Official Review · Reviewer_yHkP · 2025-03-14

**Overall Recommendation:** 3

**Summary:**

The paper introduces Griffin, the first graph-centric foundation model designed specifically for relational databases. Griffin combines advanced architectural innovations such as unified encoders for categorical and numerical features, cross-attention modules for selective information aggregation, and enhanced hierarchical message-passing neural networks. Pretrained on extensive single-table and multi-table datasets, Griffin demonstrates superior or comparable performance to task-specific models, excels particularly in low-data scenarios, and achieves strong transferability across various RDB tasks.

## No score updates after rebuttal
The comparison with single-table methods using single table input is critical from my point of view, but the rebuttal has not provided statistical results.

**Claims And Evidence:**

The claims presented in the paper are supported by extensive experimental evidence, clearly demonstrating the model's efficacy in handling relational databases. Griffin’s generalization capabilities are well-substantiated through comprehensive experiments involving benchmarks such as 4DBInfer and RelBench. However, an area that warrants further exploration is whether the method effectively generalizes to single-table datasets, especially compared to established single-table baselines.

**Essential References Not Discussed:**

-

**Experimental Designs Or Analyses:**

The experimental designs and analyses appear sound and thorough. The paper includes detailed comparisons across multiple metrics (Accuracy, ROC-AUC, RMSE, MAE, and Logloss). One limitation, however, is the restricted baseline comparison primarily against SAGE. Including comparisons with additional baseline methods tailored for single-table data would strengthen the analysis.

**Methods And Evaluation Criteria:**

The proposed methods, including unified feature encoding, cross-attention mechanisms, and hierarchical message-passing neural networks, are highly relevant and thoughtfully designed for the targeted relational database applications. The evaluation criteria using existing graph-centric benchmarks (4DBInfer and RelBench) are appropriate and rigorous, making sense for the context of relational data.

**Other Comments Or Suggestions:**

-

**Other Strengths And Weaknesses:**

Strengths:

Methodological innovations for relational database-specific foundation model.

Extensive empirical analysis, including comprehensive ablation studies and transferability experiments.

Weaknesses:

Potential for further clarification on generalizability specifically in single-table scenarios.

**Questions For Authors:**

How does Griffin perform specifically in single-table settings compared to state-of-the-art single-table models (e.g., XGBoost, TabNet, TabLLM, TableGPT)? Clarifying this would enhance understanding of its generalizability.

What is Griffin’s performance trend with varying numbers of tables in relational datasets? Is there a threshold where Griffin clearly outperforms simpler methods?

**Relation To Broader Scientific Literature:**

The integration of advanced encoding and decoding strategies from related literature on tabular and graph data enriches the paper’s contributions.

**Theoretical Claims:**

The paper does not primarily focus on theoretical claims or proofs; therefore, no correctness of proofs needed verification.

---

> ### Author Rebuttal · Authors · 2025-04-01
>
> We thank Reviewer yHkP for acknowledging the contributions of our work and for the constructive questions and suggestions. The updated experiments including 7 additional baselines about main experiment and 2 additional baselines on few-shot setting is provided at https://anonymous.4open.science/r/Griffin-Rebuttal. We address the key concerns below:
>
> ---
>
> **Q1: Performance Comparison with Single-Table Baselines**
>
> Griffin is primarily designed for RDB tasks and is not optimized to outperform state-of-the-art models on purely single-table datasets. However, to better understand its generalizability—and in response to suggestions from other reviewers—we conducted two additional sets of experiments:
>
> 1.	**DFS + Single-Table Models on RDB Tasks**:
>
> We employed **Deep Feature Synthesis (DFS)** [1], which converts multi-table RDBs into single-table formats by aggregating features. This allows single-table models to be applied to RDB tasks. Building on pipelines from 4DBInfer, we evaluated advanced single-table baselines (including MLP, DeepFM, Feature Transformer, and XGBoost) on RelBench using DFS. Results indicate that even strong single-table models struggle to match Griffin’s performance, particularly on tasks where the multi-table structure plays a critical role.
>
>
> 2.	**Few-Shot Settings with Pretrained Single-Table Models**:
>
> To evaluate performance in few-shot scenarios, we introduced two representative baselines: **TabPFN** (trained from scratch) and **TabLLM** (leveraging LLM-based pretraining). Both are designed for few-shot tabular tasks. Griffin still achieves higher average performance under this setting, demonstrating its generalization ability and potential as a foundation model for RDBs.
>
> [1] Deep Feature Synthesis: Towards Automating Data Science Endeavors https://groups.csail.mit.edu/EVO-DesignOpt/groupWebSite/uploads/Site/DSAA_DSM_2015.pdf
>
> ---
>
> **Q2: Performance Trends with Varying Numbers of Tables**
>
> Thank you for the insightful question. Griffin’s framework integrates innovations across encoder/decoder design, GNN architecture, and pretraining strategies—each contributing to improved performance under different structural complexities. Our ablation studies isolate the effect of these components, including cross-attention mechanisms and hierarchical aggregation functions.
>
> Regarding the trend with varying numbers of tables, we do not yet have a fully consistent explanation for all scenarios where Griffin outperforms simpler models. One intuitive observation is that Griffin performs especially well in tasks that involve **high-quality textual features**—for example, predicting review ratings in e-commerce datasets where review content is informative.
>
> That said, there are still cases where Griffin and simpler models (e.g., SAGE) exhibit unexpected performance patterns. Interestingly, even more expressive GNN models such as GAT, PNA, and HGT occasionally perform worse than SAGE. We believe this may stem from data-specific factors, including noisy features, varying relation types, or mismatched inductive biases—cases where simpler architectures might generalize better.
>
> We acknowledge this as an open research question and welcome further discussion on better understanding the relationship between RDB schema complexity and model performance.

---

### Official Review · Reviewer_w5kU · 2025-03-22

**Overall Recommendation:** 3

**Summary:**

The proposes Griffin, a pretrained model for relational databases. Griffin uses concepts of unified representation of inputs and tasks, cross-attention mechanisms, and graph neural networks (or MPNNs), tasks of cell completion and supervised learning for pretraining. The proposed framework was tested on several datasets of relational database and shows competitiveness compared to baselines.

**Claims And Evidence:**

Please refer to the comments or suggestions.

**Essential References Not Discussed:**

- TabPFN models in foundation models for tabular data.
- Some works on relational data (rdf2vec, EmbDI).

**Experimental Designs Or Analyses:**

Please refer to the comments or suggestions.

**Methods And Evaluation Criteria:**

Please refer to the comments or suggestions.

**Other Comments Or Suggestions:**

- What does Griffin stand for?
- I am uncertain as Griffin can be considered as a "foundation model", as it is trained on specific sets of data. In this sense, the word "towards"  in the title to me seems more acceptable and the claim (in introduction) that Griffin is a foundation model designed for RDBs seems debatable.
- The sentence "curated a diverse and extensive collection of datasets for both single-table and RDB tasks" seems to obscure. What are some of the groundwork (good efforts) that have been put to curate these data?
- Throughout the paper, the paper uses many adverbs or adjectives that are unclear. For instance, in Figure 1, what does "seamless" mean? There are several cases in the paper that use such words without much justification.
- The term (or the title "Task Unification" (3.1)) might be misleading. The term task could simply denote regression or classification, while the subsection also deals with unification of the input representation. A clarification would help the understanding.
- I am a bit confused with the part "Categorical and Textual Feature" with the paragraph starting with "For classfication tasks...". What is this supposed to describe?
- For numerical features, have the authors considered numerical embeddings in RealMLP or followups of Gorishniy et al., 2021?
- What is the structure of ENC (the numerical encoder) in equation 2?
- How does Griffin mask for the completion task? Does it have a special embedding for this mask?
- What is the number of epochs (or number of steps) for the pretraining (especially for the mask completion)?
- Since there is no mention in the number of epochs (or steps), except for employing early stopping at 10, I worry about the overfit of the pretrain of mask completion since it is based on similarity of vectors (eq. 6), and there seems to be no regularization on this (as far as the current content of the paper).
- It would be good to see comparison with simple baselines, such as DFS + (GBDT or Linear models).
- (Paragraph Message-Passing Neural Network) What defines a more stable model? Or rather what is less stable model?
- From my understanding I think there is no pretraining involved. If this is the case, it might be good to state this (hinting that it is to test the architectural choices of Griffin).
- In Figure 2, it would be really helpful to have an arrow in the metrics to point the direction of better performing. Also, the terms Griffin-Pretrain and Griffin is confusing since I feel Griffin itself is a framework that includes the pretraining. Possibly, without pretrain could be better for understanding.

**Other Strengths And Weaknesses:**

Please refer to the comments or suggestions.

**Questions For Authors:**

Please refer to the comments or suggestions.

**Relation To Broader Scientific Literature:**

Relational tabular data are widespread and some parts of the proposed methods can be extended to domain specific problems. Moreover, some techniques (e.g., cell completion) can be extended and used for pretrained neural network models for tabular data in general.

**Theoretical Claims:**

Please refer to the comments or suggestions.

---

> ### Author Rebuttal · Authors · 2025-04-01
>
> We thank Reviewer w5kU for the careful reading and detailed feedback. We address the concerns and questions below. The updated experiments including 7 additional baselines about main experiment and 2 additional baselines on few-shot setting is provided at https://anonymous.4open.science/r/Griffin-Rebuttal
>
> ---
>
> **Related Work**
>
> Thank you for the suggestions. We have included those references and also in comparison. The updated can be refered to reply to Reviewer 95m6.
>
> ---
>
> **C1: What does Griffin stand for?**
>
> Griffin stands for towards a **G**raph-centric Relat**I**onal dataset **F**oundat**I**o**N** model.
>
> ---
>
> **C2: Clarifying “Foundation Model” Usage**
>
> We appreciate your point. While Griffin is trained on specific datasets, we consider it to follow the general trajectory of foundation models [1] —namely, training on broad data for wide applicability. That said, we agree the claim should be stated with more care. We now consistently refer to Griffin as “**towards** a foundation model” or as“ a foundation model **attemptation**”. All inconsistent uses have been revised accordingly.
>
> [1] On the Opportunities and Risks of Foundation Models
>
> ---
>
> **C3: Clarifying Data Curation Efforts**
>
> Thank you for raising this. The data curation involved collecting, filtering, and improving dataset quality. We started with public datasets from TPBERTa, TabLLM, UniTabE, TableGPT, TaBERT, TabNet, CARTE, and others from HuggingFace. However, many were unsuitable—some tables had very few rows, others were derived from RDBs but had lost context when flattened, and some were noisy or irrelevant (e.g., meaningless string columns without clear semantics).
>
> We ultimately curated ~200 datasets with over 10M rows. These were selected for having rich metadata and column features not heavily preprocessed (e.g., not just hashed values). We also generated natural language task descriptions using GPT4 to improve the training signals for tasks.
>
> ---
>
> **C4 & C13: Ambiguous Adjectives**
>
> We appreciate this feedback and have revised the paper to remove or clarify ambiguous adjectives. Specifically:
>
> •	The word **“seamless”** was used to describe the unified modeling of both single-table and RDBs as graphs using a consistent node structure.
>
> •	The term **“stable”** referred to the aggregation mechanism in our heterogeneous GNN, where neighbors are first aggregated within each relation type, followed by max aggregation across types to focus on the most relevant relations. Ablations also verify the effectiveness.
>
> For them, we have now replaced the adjective with a direct explanation of the method.
>
> ---
>
> **C5 & C6: Clarifying Section 3.1 and Decoder Design**
>
> We revised Section 3.1 to clarify the notion of **“Task Unification”**. Previously, it refers not only to regression/classification unification but also to input unification. Now we split the subsection into two subparts: one describing encoder design and one describing decoder design. For classification tasks, the decoder retrieves the most similar label embedding. These embeddings are aligned with both category and textual feature encodings. For regression tasks, a numerical decoder predicts normalized values.
>
> ---
>
> **C7 & C8: Numerical Feature Encoder**
>
> The ENC module is a 3-layer MLP with SiLU activations (replacing ReLU) and layer normalization. Our current pretraining task about ENC and DEC focuses on accurate numerical recovery, for which this design is sufficient. We included this paper in related work and leave further exploration on numbers to future work.
>
> ---
>
> **C9: Masking Strategy for Completion Task**
>
> We use a simple zero-filling strategy, consistent with the settings used in 4DBInfer and RelBench. Zero is also used as the placeholder during prediction.
>
> ---
>
> **C10–C11 & C14–C17: Pretraining Details**
>
> Thank you for your careful reading. We now provide a detailed breakdown of the training regimes used in our experiments:
>
> | **Method** | **Completion-pretrain-single** | **Joint-SFT-single** | **Joint-SFT-RDB** | **Finetune** |
> | --- | --- | --- | --- | --- |
> | **Data volume** | ~10M rows | ~1M rows | ~150M total (domain-specific subset) | Task-specific |
> | **Training steps** | ~12k (batch size 4096) | ~6k (batch size 4096) | Domain-dependent | Task-dependent |
> | **Regularization** | L2 (9e-3) | L2 (2e-5) | L2 (2e-4) | L2 (2e-4) |
>
>
> To improve clarity in the presentation of our experiments, we have renamed the models as follows:
>
> •	**Griffin-unpretrained**: trained from scratch, no pretraining
>
> •	**Griffin-pretrained**: pretrained on single-table data (completion + joint SFT)
>
> •	**Griffin-RDB-SFT**: further pretrained with RDB-based joint SFT
>
>
> ---
>
> **C12: DFS + X Baselines**
>
> We have added these baselines in our updated experiments. A summary of the updated experimental setup is provided in our response to Reviewer zrC4 under Evaluation Concerns.

---

### Official Review · Reviewer_zrC4 · 2025-03-24

**Overall Recommendation:** 2

**Summary:**

This paper introduces Griffin, which is claimed to be a novel foundation model specifically designed for RDBs. Griffin aims to unify differet tasks from single table to multi table RDBs. To do that, Griffin is pretrained by sampling the sub graphs from RDBs, and use a unified encoder/decoder to generate unified embeddings for different tasks. Experiments shows that Griffin has good performance across different tasks.

**Claims And Evidence:**

My major concerns of this paper are (1) the novelty of the proposed approach and (2) the insufficient discussion and comparison with other related works. For the novelty, the major contribution claimed was the introduction of a unified task decoder that eliminates the need for different prediction heads for different tasks. However, this idea seems to be not very novel or even needed. Instead of using the prediction heads for different tasks, Griffin still needs an MLP to be trained by sampling "x". It seems like simply merging all different prediction heads into one head. I'd suggest the authors to clarify the novelty of this paper.

For the discussion of related works, see section "Methods And Evaluation Criteria".

**Essential References Not Discussed:**

There are multiple methods not mentioned in this paper, which are very relavent to this domain. For example, this paper is also similar to those that talks about embedding over relational table (eg. https://arxiv.org/pdf/1909.01120). Some other LLM methods for tabular questions are also worth discussing: eg. https://arxiv.org/abs/2004.02349  https://arxiv.org/pdf/2107.07653  https://arxiv.org/pdf/2207.03637

**Experimental Designs Or Analyses:**

See "Methods And Evaluation Criteria".

**Methods And Evaluation Criteria:**

In the experiment section, the authors focuses on the comparison with only one baseline approach SAGE. I understand that the authors may want to do a "fair" comparison among different method. However, it is still very necessary to include other recent baselines into the evaluation, such as the ones reported in 4DBInfer or whichever performs well on the reported benchmarks (Figure 2). The authors mentioned that they did some modification due to the normalization etc. IMO, this should not be a reason not to include more baseline result. From the Appendix b.2, the results of Griffin can be transformed back so I did not see a challenge there.

**Other Comments Or Suggestions:**

See above

**Other Strengths And Weaknesses:**

W1: I think some part of this paper is not well-written and handwavy. E.g. for appendix B.1, I think the authors want to talk about the experimental setup difference between the methods in 4DBInfer RelBench and this paper. But is it confusing to say comprision between a benchmark with a model (griffin). E.g. Figure 1 shows a subgraph sampling phase but seems this phase is not described in the methodology section. Also figure 2 also shows a very good running example. I'd suggest the authors use this example in Section 3 while you explain your workflow.

W2: Lack explaination of the results. In figure 2, it is intersting to show that on task "Retailrocket/cvr", Griffin is significantly better than SAGE. But for other benchmarks, the gap is much smaller, even on some benchmarks Griffin is worse. These results are interesting and need some explaination.

**Questions For Authors:**

Could you please clarify the concerns that I have in section "Claims And Evidence" and "Methods And Evaluation Criteria"?

**Relation To Broader Scientific Literature:**

This paper introdcues some new ideas to the area but like I mentioned before, I think the novelty of this paper is limited.

**Theoretical Claims:**

Not applicable.

---

> ### Author Rebuttal · Authors · 2025-04-01
>
> We thank reviewer zrC4 for the detailed and constructive feedback. Below, we address each of the main concerns. The updated experiments including 7 additional baselines about main experiment and 2 additional baselines on few-shot setting is provided at https://anonymous.4open.science/r/Griffin-Rebuttal
>
> ---
>
> **Claim 1: Novelty of the Proposed Approach**
>
> We appreciate the reviewer’s concern about the novelty of our approach and would like to clarify our contributions. Our goal is to take a step toward building a foundation model RDBs. We focus on three key challenges:
>
> 1.	How to represent different RDBs in a unified model?
>
> 2.	How to design a GNN architecture that works effectively for RDBs?
>
> 3.	How to leverage abundant data, and when does it help?
>
> To address these challenges, we propose **Griffin**, a unified framework with three main components: (1) encoder/decoder design, (2) GNN architecture updates, and (3) a pretraining pipeline. Then our experiments show consistent improvement through three stages:
>
> 1.	Our base model (even without pretraining) outperforms prior models (initially compared only with SAGE; we have now added 7 more baselines).
>
> 2.	Pretraining on single-table data further boosts performance.
>
> 3.	Additional SFT with similar or diverse RDBs enhances performance, especially in low-resource settings.
>
> The strength of **Griffin** lies in the integration of these three components under a unified framework aimed at RDB foundation modeling. Each component plays a distinct and necessary role: enabling diverse data usage, improving learning capacity, and supporting practical deployment. Our experiments confirm that each part contributes meaningfully to both performance and transferability.
>
> ---
>
> **Claim 2: Discussion of Related Work**
>
> We thank the reviewer for highlighting missing related work. We have now revised the related work section to include a wider range of relevant literature. The updated can be refered to reply to Reviewer 95m6.
>
>
> ---
>
> **Evaluation Concern: Missing Baselines and Alignment with RelBench**
>
> Thank you for raising this important point. While we initially prioritized “fair” comparisons, we agree that broader baseline coverage is necessary for a more complete evaluation. In response, we have expanded our baseline set to include:
>
> •	**Three additional GNN-based baselines**: GAT, PNA, and HGT, as suggested in 4DBInfer.
>
> •	**Four single-table baselines**: MLP, DeepFM, Feature Transformer, and XGBoost, along with **Deep Feature Synthesis (DFS)**—a strong feature synthesis method that converts multi-table data into a single table by computing meaningful feature combinations.
>
> •	**Two single-table baselines for few-shot settings**: We include TabPFN and TabLLM as representative few-shot baselines, both of which leverage pretrained models. TabPFN is trained from scratch, while TabLLM uses LLM-based pretraining. Although DFS can be computationally intensive (taking up to 7 hours for optimized pipelines in 4DBInfer and even longer using the original Featuretools implementation), making it less suitable for low-resource few-shot settings, we still include these baselines as a reference.
>
> For evaluation, we continue to present normalized scores in the main figures to support comparison across tasks with varying scales. To ensure transparency and enable alignment with RelBench, we also provide some unnormalized results which is aligned with normalized scores, both suggesting Griffin outperforms Sage.
>
> ---
>
> **W1: Method Description and Figures**
>
> We thank the reviewer for this helpful feedback. We have made several improvements including Clarified terminology and Improved explanation of subgraph sampling and method pipeline.
>
> ---
>
> **W2: Explanation of Results**
>
> Thank you for the suggestion to provide more explanation of the results. In general, Griffin performs better when the task benefits from rich table metadata and high-quality text feature embeddings.
>
> However, there are still cases where it is difficult to consistently explain why Griffin outperforms SAGE, or vice versa. This challenge also applies to other strong GNN baselines such as GAT, PNA, and HGT. Despite being more expressive in theory, these models sometimes perform significantly worse than SAGE. We believe this may be due to the complexity and variability in data distributions, where simpler models like SAGE may align better with the task’s inductive bias in certain cases.
>
> We acknowledge this as an open question and an area for future work. We welcome further insights and suggestions on how to better understand and interpret these variations.

---

> > ### Comment · Reviewer_zrC4 · 2025-04-03
> >
> > Thanks for the responses. I've read them and I'll update my score if necessary.

---

> > > ### Author Response · Authors · 2025-04-03
> > >
> > > Thank you for your reply! Your comments are very valuable to us and have helped improve the quality of our work. We would be happy if our responses addressed your concerns and if you would consider raising your score. The experiments and revisions based on your feedback, as well as suggestions from other reviewers, will be carefully included in the final version of the paper.

---

### Decision · Program_Chairs · 2025-05-01

**Decision:**

Accept (poster)

**Comment:**

The paper proposes a novel framework for pre-trained models on relational tabular data. While the original submission contained only limited related work, the authors provided extensive comparison to other models on relational data in the rebuttal, showing a clear advantage for the proposed method. However, the exact selection criteria for the datasets that were included are not provided, potentially raising a methodological concern.